# Atmospheric oxidation of 1,3-butadiene: influence of seed aerosol acidity and relative humidity on SOA composition and the production of air toxic compounds

Mohammed Jaoui[1], Klara Nestorowicz[2,5], Krzysztof Jan Rudzinski[2], Michael Lewandowski[1], Tadeusz E. Kleindienst[1], Julio Torres[3], Ewa Bulska[3], Witold Danikiewicz[4] and Rafal Szmigielski[2]

[1]Center for Environmental Measurement & Modeling, U.S. Environmental Protection Agency, Research Triangle Park, NC, 27711, USA.
[2]Environmental Chemistry Group, Institute of Physical Chemistry Polish Academy of Sciences, 01-224 Warsaw, Poland
[3]University of Warsaw, Faculty of Chemistry, Biological and Chemical Research Centre, 02-089 Warsaw, Zwirki i Wigury 101, Poland.
[4]Mass Spectrometry Group, Institute of Organic Chemistry, Polish Academy of Science, 01-224 Warsaw, Poland
[5]Mass Spectrometry Laboratory, Institute of Organic Chemistry, Polish Academy of Science, 01-224 Warsaw, Poland

*Correspondence to*: Mohammed Jaoui (jaoui.mohammed@epa.gov); Rafal Szmigielski (ralf@ichf.edu.pl)

**Abstract.** This study investigated the effect of relative humidity (RH) on the chemical composition of gas and particle phases formed from the photooxidation of 1,3-butadiene (13BD) in the presence of NOx under acidified and non-acidified seed aerosol. The experiments were conducted in a 14.5 m$^3$ smog chamber operated in a steady-state mode. Products were identified by high performance liquid chromatography, gas chromatography mass spectrometry and ultrahigh performance liquid chromatography coupled with high resolution mass spectrometry. More than 50 oxygenated products were identified including 33 oxygenated organics, 10 organosulfates (OSs), PAN, APAN, glyoxal, formaldehyde, and acrolein. Secondary organic aerosol (SOA) mass and reaction products formed depended on RH, and acidity of the seed aerosol. Based on E-AIM modeling, the seed aerosol originated from the acidified and non-acidified solutions were found to exist under aqueous and solid phases, respectively. Although, the terms "acidified" and "non-acidified" is true for the solutions from which the seeds were atomized, there is far more fundamental differences between the phase states in which species partition to or from (aqueous/solid), which considerably affects their partitioning and formation mechanisms. SOA mass, and most SOA products (i) were higher under acidified seed conditions, where the aerosol particles were deliquescent, than under non-acidified seed conditions, where the aerosol particles did not contain any aqueous phase, (ii) increased with the acidity of the aerosol aqueous phase in the experiments under acidified seed conditions; and (iii) decreased with increasing RH. Glyceric acid, threitols, threonic acids, four dimers, three unknowns, and four organosulfates were among the main species measured either under acidified or non-acidified conditions across all RH levels. Total secondary organic carbon and carbon yield decreased with increasing RH under both acidified and non-acidified seed conditions. The photochemical reactivity of 13BD in our systems decreased with increasing RH and was faster under non-acidified than acidified seed conditions. To determine the contribution of 13BD products to ambient aerosol, we analyzed PM$_{2.5}$ samples collected at three European monitoring stations located in Poland. The occurrence of several 13BD SOA products (e.g., glyceric acid, tartronic acid, threonic acid, tartaric acid, and OSs) in the field samples suggests that 13BD could contribute to ambient aerosol formation.

**Keywords:** 1,3-butadiene, SOA, formaldehyde, acrolein, HAPs, relative humidity, organosulfates, silylation

# 1. Introduction

Atmospheric particulate matter, especially particles with diameters less than 2.5 micrometres (PM$_{2.5}$), plays an important role in Earth's climate through direct scattering and absorption of incoming solar radiation, trapping of outgoing long-wave radiation (Charlson et al. 1992), and altering cloud formation and lifetime (Shrivastava et al., 2017; Flecher et al., 2018). PM$_{2.5}$ is considered harmful to humans and the environment (Majewska et al., 2021; Manisalidis et al., 2020) but its impact on climate, air quality, health, and the chemistry of the atmosphere is still not adequately understood (Myhre et al., 2013, Noziere et al, 2015). Recent literature suggests that secondary organic aerosol (SOA) contributes significantly to ambient PM$_{2.5}$, and remains a relevant subject of research (Jimenez et al., 2009; Shrivastava et al., 2017; Srivastava et al., 2022). A substantial number of studies have been conducted to investigate SOA chemical composition, formation mechanisms and contribution to ambient PM$_{2.5}$. SOA is formed through the oxidation reactions of volatile organic compounds (VOCs) that are emitted directly into the atmosphere from anthropogenic and biogenic sources, producing highly oxidized species of low volatility, which can form new particles or condense onto existing particle surfaces. Precursor VOCs emitted in urban and industrial areas have significant influence on aerosol burden either through direct SOA formation or enhancement of SOA originated from naturally emitted precursors (Jaoui et al., 2008; Hoyle et al., 2011).

The mechanism of SOA formation is complex and depends on several environmental parameters including temperature, relative humidity (RH), acidity of the aqueous particles, solar irradiance, type of oxidant, and the concentration of precursor VOCs that controls peroxy radicals (RO$_2$) and NOx concentrations. All these parameters can affect SOA chemical composition, an important characteristic of atmospheric particles that may provide key insights on properties governing the effect of particles on climate, air quality, and human health. While considerable effort has been expended studying acidity and RH effects on biogenic precursors, far less effort has been made to examine such effects on anthropogenic hydrocarbons including reactive hazardous air pollutant (HAP) species, such as 1,3-butadiene (13BD) and benzene. RH affects the quantity of water available in the system, by influencing the processes in which water acts as a reactant, catalyst, product, or solvent. Atmospheric water also has a strong effect on the acidity of aqueous particles (Tilgner et al., 2021; Lei et al., 2022; Cook et al., 2024). While the effect of RH on biogenic and anthropogenic SOA bulk properties (e.g., SOA yield) has been the subject of a number of studies (Edney et al., 2000; Cocker III et al., 2001; Zhou et al., 2011; Nguyen et al., 2011; Zhang et al., 2011; Kamens et al., 2011; Lewandowski et al., 2015; Riva et al., 2016; Faust et al., 2017; Hinks et al., 1918; Liu et al., 2019a; Healy et al., 2009; Jia and Xu, 2014, 2018; Chen et al., 2021), the effect of RH on SOA chemical composition is complex and has been reported only in a few studies (Nestorowicz et al. 2018; Wang et al. 2020; Zhang et al., 2021; Klodt et al., 2023; Luo et al., 2024; Thomsen et al., 2024).

Although significant progress has been made over the past three decades in improving our understanding of the atmospheric chemistry controlling the formation and transformation of air pollutants, significant uncertainties remain as the suite of chemical compounds involved is ever expanding. HAPs, also known as air toxics, are an increasingly studied group of chemicals because they are known to be potentially carcinogenic and dangerous for human health and the environment (Scheffe et al., 2016; Kao et al., 1994). HAPs are either emitted directly (primary HAPs or PHAPs) or formed through secondary reactions in the atmosphere (SHAPs). For example, acrolein, formaldehyde, and acetaldehyde are important HAPs from a health and atmospheric chemistry standpoint, yet their sources and reactions remain uncertain. Regulatory policies for HAPs target the sources of direct emissions, however, several aromatic, nitrogenated, and oxygenated HAPs can be formed in the atmosphere through chemical reactions. Numerous PHAPs undergo atmospheric transformation following their emissions by reacting with atmospheric oxidants, leading to the formation of SOA and SHAPs. For example, formaldehyde is produced from almost every atmospheric photooxidation reaction (Kao et al., 1994).

1,3-Butadiene, a structural analog of isoprene, is an important anthropogenic PHAP released into the atmosphere through industrial petroleum processing for synthetic rubber, resins and plastics production, combustion of vehicle fuels, tobacco smoke and biomass combustion (Berndt and Böge, 2007, Anttinen-Klemetti et al., 2006; Dollard et al., 2001; Eatough et al., 1990; Pankow et al., 2004; Penn and Snyder, 1996; Ye et al., 1998; Thornton-Manning et al., 1997; Sorsa et al., 1996; Hurst, 2007). 13BD is classified as a hazardous (U.S. EPA, 1996), carcinogenic, toxic, mutagenic, and genotoxic pollutant in humans and other mammals (Acquavella, 1996; IARC, 2012; IPCS, 2001; US EPA, 2002). The annual emission of 13BD into the atmosphere is estimated at six million tons worldwide (Berndt and Böge, 2007). A recent paper by Chen and Zhang (2022) provides an update on the properties of 13BD from newly available data. 13BD reacts in the atmosphere with OH, O$_3$, NO$_3$ and Cl and can produce many potential toxic products including acrolein and formaldehyde (U.S. EPA, 1996; Angove et al.,

2006; Doyle et al., 2004; Notario et al., 1997; Kramp and Paulson, 2000; Liu et al., 1999). Previous research efforts have made significant advances in 13BD oxidation product characterization and their role in SOA formation (Angove et al., 2006; Berndt and Böge, 2007; Kramp and Paulson, 2000; Liu et al. 1999; Sato, 2008; Sato et al., 2011; Jaoui et al., 2014), however, 13BD SOA composition has not been adequately characterized and remains largely unknown. To date, few smog chamber studies have been focused on 13BD SOA formation (Angove at al., 2006; Sato, 2008; Sato et al., 2011; Jaoui et al., 2014), and only a limited number has been focused on the effects of seed aerosol acidity and relative humidity on SOA formation (Lewandowski et al. 2015). Our previous work (Jaoui et al. 2014) reported a detailed chemical characterization of gas and particle phases formed from the oxidation of 13BD in the presence (RH = 30%) and absence (RH < 3%: dry conditions) of NOx. We identified several oxygenated organic compounds identified using GC-MS analysis, including threitol, erythritol, glyceric acid, malic acid, tartaric acid, threonic acid. Many of these compounds were detected in ambient aerosol. Our previous work (Jaoui et al., 2014) focused on non-sulfate species analysis due to the capabilities of the analytical method used, therefore additional characterization of SOA from 13BD oxidation is still needed to characterize organosulfates (OSs) and nitro- or nitroso-organosulfates (NOSs).

The objective of the present study is to assess the impact of aerosol water content on acidity-influenced SOA formation during NOx -mediated irradiation of 13BD. It extends our work previously reported showing the critical role RH plays in SOA composition from the irradiated isoprene/NOx system (Lewandowski et al., 2015; Nestorowicz et al., 2018). Our previous work focused mainly on the role of RH (< 5% and 30%) on aerosol bulk parameters, that is, secondary organic carbon (SOC) yields and OM/OC (Lewandowski et al., 2015; Jaoui et al., 2014). Here we explored the effect of RH on a wide range of 13BD oxygenated products in the presence of acidified and non-acidified sulfate seed aerosol and incorporated a new method to analyse organosulfates (OSs) using a liquid chromatography–mass spectrometry (LC-MS) technique (Szmigielski, 2016; Rudzinski et al., 2009; Darer et al., 2011). Non-sulfate SOA oxygenates were also examined using silylation derivatization followed by GC-MS analysis (Jaoui et al., 2004). We also analyzed the influence of RH and acidity of seed aerosol on HAPs formation, as well as glyoxal, peroxyacetyl nitrate (PAN) and peroxyacryloyl nitrate (APAN). The results were compared to the chemical analysis of $PM_{2.5}$ field samples to substantiate the importance of the laboratory findings and their role in ambient SOA formation. Numerical calculations using the Extended Aerosol Inorganic Model (E-AIM) were used to examine the response of RH in aerosols upon the addition of sulfuric acid. The results of the present study contribute to further development of acidity-influenced SOA chemistry in air quality models.

## 2. Experimental methods

All chemicals used in the smog chamber experiments and GC-MS analysis, including 13BD, external standards used for calibration curves, derivatization agent N,O-bis(trimethylsilyl)trifluoroacetamide (BSTFA) with 1% of trimethylchlorosilane (TMCS) as catalyst, were purchased from Aldrich Chemical Co. (Milwaukee, WI, USA) at the highest purity available and were used without further purification. Solvents with $GC^2$ quality were purchased from Burdick and Jackson (Muskegon, MI, USA). Standard compounds for LC-MS analysis –DL-tartaric acid (99%), D-threono-lactone (≥95%), tartronic acid (>97%), DL-malic acid (≥99%) and sodium 1-pentyl sulfate (99%) were purchased from Sigma Aldrich, Poland at the highest available purity and used without further purification. Aerosol extracts and LC-MS mobile phases were prepared using high-purity water (resistivity 18.2 MΩ·cm$^{-1}$) from a Milli-Q Advantage water purification system (Merck, Poland) and high purity methanol (LC-MS ChromaSolv-Grade) were purchased from Sigma-Aldrich, Poland. D-Threonic acid was prepared in our laboratory according to the procedure described in Jaoui et al. (2019).

### 2.1 Smog chamber experiments

The experiments were conducted in a 14.5 m$^3$ fixed-volume chamber with a stainless-steel frame and interior walls fused with a 40 μm PTFE Teflon coating. The chamber operation, sample collection, derivatization procedure, and the GC-MS and LC-MS analysis methods used in this study has been described in detail elsewhere (Jaoui et al., 2004, 2014; Kleindienst et al., 2006; Nestorowicz et al., 2018). The chamber was operated in flow mode as a continuous stirred tank reactor with a residence time of 4 h to produce a steady-state, constant aerosol distribution which could be repeatedly sampled at various seed aerosol

acidities and relative humidities. A set of UV fluorescent bulbs was used in the chamber to produce radiation in the actinic region of the spectrum at 300-400 nm photolytically comparable to that of solar radiation (Black et al., 1998).

13BD and NO were added continuously from high-pressure cylinders, each diluted with $N_2$, through flow controllers into the inlet manifold, where they were diluted and mixed prior to the introduction to the chamber. Inorganic seed aerosol was added continuously to the chamber by nebulizing (TSI, model 9302, Shoreville, MN) dilute aqueous solution of ammonium sulfate (non-acidified seed aerosol) or aqueous solution of ammonium sulfate and sulfuric acid (acidified seed aerosol) with the total sulfate concentration held constant to maintain stable inorganic concentrations in the chamber (Lewandowski et al., 2015). The seed aerosol stream was equilibrated to the computer-controlled relative humidity designated for a particular experiment. The reaction run for about 24 hours to reach the steady-state conditions. The concentrations of 13BD in the inlet manifold and the chamber were measured using a gas chromatograph with flame ionization detection (Hewlett-Packard, Model 5890 GC). NO and NOx were measured with a chemiluminescent analyzer (TECO, Model 42C, Franklin, MA), and $O_3$ was measured with a chemiluminescent ozone monitor (Bendix, Model 8002, Lewisburg, WV). Temperature and relative humidity were measured with a thermo-hydrometer (Omega Engineering, Inc., Model RH411, Stamford, CT). The effect of relative humidity on 13BD SOA composition was examined with two sets of experiments, each with multiple stages. In the first set of experiments, 13BD/NO system was examined at multiple humidity levels using a low concentration (1 µg m$^{-3}$) of ammonium sulfate seed aerosol (ER666, Table 1). The relative humidity was varied in stages from roughly 11 to 60 % (Table 1). The overall average temperature was ~25 °C. This set provided a base case for exploring the changes in aerosol composition in the absence of significant seed aerosol acidity. In the second set of experiments, the same system 13BD/NO was examined in the presence of an acidified inorganic sulfate seed aerosol, generated using a solution incorporating sulfuric acid solution and ammonium sulfate solution (Table 1), in which sulfate concentration was held constant across the full range of humidity examined. Additional bulk aerosol analysis associated with these experiments were reported by Lewandowski et al. (2015). Secondary organic aerosol produced in the chamber at each stage was collected for 24 h at 16.7 L min$^{-1}$ on 47 mm Glass fiber filters (GF) (Pall Corporation, Ann Arbor, MI).

**Table 1.** 13BD photooxidation smog chamber experiments in the presence of acidified and non-acidified seed aerosol – initial and steady state conditions, reacted and 13BD conversion and yields data. The initial NOx was entirely nitric oxide. The non-acidified experiments were conducted with ammonium sulfate seed at ~1 µg m$^{-3}$. The acidified experiments were conducted with inorganic seed generated from a nebulized solution for which two third the sulfate mass was derived from sulfuric acid and the other one third from ammonium sulfate, giving a constant aerosol sulfate concentration at ~35 µg m$^{-3}$ (Lewandowski et al., 2015). For a wide range of particles, the wall loss rate is 0.063 h$^{-1}$. However, with the chamber running in a flow mode the wall loss rate is subsumed in the observed decrease from the input reactants and the steady-state concentrations. The steady state SOCs for both experiments are wall-loss corrected. *: NO explicit measurement specific to that stage is reported; and for the remaining stages, a full experiment (all stages) global NO average is reported. The conversion efficiency is defined as in Liu et al. (2019b). For filters collected and numbering see section 2.3 below. [H$^+$] air were reported for ER444 (Lewandowski et al., 2015).

| Experiment ER444: Acidified seed aerosol (1/3 ammonium sulfate, 2/3 sulfuric acid by sulfate mass in precursor solution) | | | | |
|---|---|---|---|---|
| | Stage 1 | Stage 2 | Stage 3 | Stage 4 |
| Initial 1,3-butadiene (ppmC) | 6.88 | 7.02 | 6.81 | 6.92 |
| Initial NO (ppm) | 0.313* | 0.343 | 0.373* | 0.343 |
| **Steady state conditions** | | | | |
| RH (%) | 10 | 31 | 50 | 62 |
| Temperature (°C) | 22.4 | 22.6 | 22.2 | 22.2 |
| $O_3$ (ppm) | 0.313 | 0.259 | 0.255 | 0.242 |
| NOy (ppm) | 0.168 | 0.150 | 0.129 | 0.117 |
| ΔHC (µg m$^{-3}$) | 2842 | 2848 | 2640 | 2611 |
| Conversion efficiency (%) | 74.1 | 72.6 | 69.6 | 67.6 |
| SOC (µgC m$^{-3}$) | 60.3 | 41.6 | 33.7 | 31.1 |
| Carbon Yield | 0.024 | 0.016 | 0.014 | 0.013 |
| Filter analyzed: LC-MS | GF21 | GF9 | GF18 | GF24 |
| GC-MS | GF19 | GF7 | GF16 | GF23 |

| Experiment ER666: Non-acidified seed aerosol (ammonium sulfate) | | | | | | |
|---|---|---|---|---|---|---|
| | Stage 1 | Stage 2 | Stage 3 | Stage 4 | Stage 5 | Stage 6 |
| Initial 1,3-butadiene (ppmC) | 7.00 | 7.00 | 7.09 | 7.14 | 7.11 | 7.13 |
| Initial NO (ppm) | 0.416 | 0.416 | 0.416 | 0.408* | 0.416 | 0.416 |
| **Steady state conditions** | | | | | | |
| RH (%) | 11 | 20 | 29 | 39 | 49 | 60 |
| Temperature (°C) | 25.3 | 25.3 | 25.1 | 24.7 | 25.0 | 24.7 |
| $O_3$ (ppm) | 0.327 | 0.314 | 0.307 | 0.290 | 0.273 | 0.254 |
| $NO_y$ (ppm) | 0.242 | 0.252 | 0.239 | 0.231 | 0.227 | 0.243 |
| $\Delta HC$ ($\mu g\ m^{-3}$) | 3099 | 3077 | 3085 | 3030 | 2968 | 2917 |
| Conversion efficiency (%) | 80.3 | 79.7 | 78.9 | 76.5 | 75.4 | 74.0 |
| SOC ($\mu gC\ m^{-3}$) | 45.1 | 40.9 | 34.7 | 27.0 | 26.4 | 24.7 |
| Carbon Yield | 0.016 | 0.015 | 0.013 | 0.010 | 0.010 | 0.010 |
| Filter analyzed: LC-MS | GF2 | GF4 | GF6 | GF8 | GF10 | GF12 |
| GC-MS | GF1 | GF3 | GF5 | GF7 | GF9 | GF11 |

## 2.2 Ambient aerosol samples

Ambient $PM_{2.5}$ samples were collected on pre-baked quartz-fiber filters with a high-volume aerosol sampler (DH-80, Digitel) at the regional background monitoring stations in (1) Zielonka (the Kuyavian-Pomeranian Province, Poland, 53°39'N, 17°55'E), (2) Diabla Góra (the Warmia-Masurian Province in Northern Poland, 54°07'N, 22°02'E) during summer 2016 campaigns, and (3) the regional monitoring station in Godów (the Silesian Province, Poland, 49°55'N 18°28'E) in summer 2014. All sites are located in forested areas, however Godów site is close to major industrial cities of the Silesian region in Poland and a coal-fired power station in Detmarovice, Czechia, so samples collected at this site were highly influenced by anthropogenic sources. Additional characteristics of Zielonka and Godów sites were reported in Nestorowicz et al. (2018). Diabla Góra is surrounded by boreal forests, and highly influenced by biogenic precursors, e.g. isoprene and α-pinene. The anthropogenic emissions are seasonal and limited to biomass burning and vehicle exhausts. Meteorological and pollution characteristics of the sampling sites are presented in Table 2.

**Table 2.** Characteristics of sampling sites in Poland: meteorological data (average temperature and RH), ambient air parameters (NOx, SO₂) and organic carbon (OC) levels in $PM_{2.5}$.

| | RH (%) | Temperature (°C) | NOx ($\mu g\ m^{-3}$) | SO₂ ($\mu g\ m^{-3}$) | OC ($\mu g\ m^{-3}$) |
|---|---|---|---|---|---|
| Zielonka | 86% | 25 | 1.3 | 0.6 | 1.7 |
| Godów | 94% | 28 | 30.0 | 3.0 | 5.4 |
| Diabla Góra | 62% | 18 | 3.2 | 0.4 | 3.3 |

## 2.3 Secondary organic aerosol analysis

**EC-OC analysis.** Organic carbon (OC) formed in each stage of the experimental studies was measured using a semi-continuous elemental carbon-organic carbon (EC-OC) instrument (Sunset Laboratories, Tigard, OR). The instrument operates with a quartz filter positioned within the oven housing used for the analysis. The pumping system drew chamber effluent through the filter at a rate of 8 L min⁻¹. A carbon-strip denuder was placed in-line before the quartz filter to remove gas-phase organic compounds in the effluent that might interfere with the OC measurement. With a sample collection time of 0.5 h and an analysis time of 0.25 h, the duty cycle for the measurement of OC was 0.75 h (Lewandowski et al., 2015).

**GC-MS analysis.** SOA formed in each stage of each experiment (Table 1) was analyzed for individual organic compounds by extracting one GF filter from each stage using sonication in a 1:1 dichloromethane/methanol mixture for 1 h. Prior to extraction, cis-ketopinic acid (KPA; 21.4 $\mu g$) and $d_{50}$-tetracosane (20.8 $\mu g$) were added to each filter as internal standards. Filter extracts were concentrated to dryness and then derivatized with *bis*(trimethylsilyl) trifluoroacetamide (BSTFA) containing 1% trimethylchorosilane as catalyst in the presence of 100 $\mu l$ pyridine. Samples were heated to 70 °C to complete the reaction (Jaoui et al., 2018). The derivatized extracts were analyzed using a ThermoQuest GC (Austin, TX, USA) coupled to an ion

trap mass spectrometer (ITMS). The injector, heated to 270 ∘C, was operated in splitless mode. Compounds were separated on a 60 m long, 0.25 mm ID RTx-5MS column (Restek, Inc., Bellefonte, PA, USA) with a 0.25 $\mu$m film thickness. The GC oven temperature program for the analysis started isothermally at 84 °C for 1 min, followed by a temperature ramp of 8 °C min$^{-1}$ to 200 °C and a 2 min hold, and was then ramped at 10 ∘C min$^{-1}$ to 300 °C. The ion source, ion trap, and interface temperatures were 200, 200, and 300 °C, respectively. Mass spectra were collected in both the methane chemical ionization (CI) and electron ionization (EI) modes (Jaoui et al., 2014). Jaoui et al. (2004) determined the efficiency of BSTFA derivatization to be greater than 95%. In addition to important pieces of information used for structural elucidation of unknown oxygenated organic species, the silylation method can be used for quantitative analysis using recovery and internal standards. Previous quantitative analysis was performed on nitroaromatic and other oxygenated organic compounds containing hydroxyl and/or carboxylic groups (Jaoui et al., 2004, 2018). In this study, calibration curves were prepared by diluting standard mixtures (Supplementary Information (SI) section S1). The compounds listed in Tables S1 and S2 have been chosen due to their structural diversity and similarity to products detected in chamber and ambient samples. The method was evaluated previously by our laboratory (Jaoui et al., 2004, 2018). Calibration curves were developed for silylated compounds as reported in Tables S1 and S2 by plotting the relative integrated peak areas versus amount relative to the internal standard (cis-ketopinic acid) as shown Figure S1. Figures S2 and S3 show examples of TIC chromatogram and mass spectra recorded for selected standards, respectively.

**LC-MS analysis (GF and ambient filters).** For LC-MS analysis, one GF filter from each stage was used as shown in Table 1. The procedure of filters extraction was similar to the one described in Nestorowicz et al. (2018). From each filter, two 1.5 cm$^2$ punches were taken, extracted with 15 mL aliquots of methanol for 60 min using a Multi-Orbital Shaker (PSU-20i, BioSan). The extracts were concentrated on rotary evaporator (Rotavapor® R215, Büchi; temperature 30ºC, vapor pressure 150 mbar), then filtered with disposable 0.2 µm PTFE syringe filters and dried to dryness under a gentle stream of nitrogen at ambient temperature. The residues were reconstituted with 200 µL of 1:1 methanol/water mixture and shaken for 1 min. The qualitative UHPLC-ESI-MS/MS analyses used were described elsewhere (Nestorowicz et al., 2018; Spolnik et al., 2018). A Waters ACQUITY UPLC I-Class chromatograph coupled with a Waters SYNAPT G2-S high-resolution mass spectrometer was used. The chromatographic separations were performed using an ACQUITY HSS T3 column (2.1×100 mm, 1.8 $\mu$m particle size) at room temperature. The mobile phases consisted of 10 mM ammonium acetate (eluent A) and methanol (eluent B). An injection volume of 1 $\mu$L was used. The SYNAPT G2-S spectrometer equipped with an electrospray ionization source (ESI) was operated in the negative-ion mode. Optimal ESI source conditions were 3 kV capillary voltage with a 20 V sampling cone and full-width-at-half-maximum mass-resolving power of 20,000. High-resolution mass spectra were recorded from $m/z$ 50 to 600 in the MS or MS/MS modes. All data were recorded and analyzed with the Waters Mass Lynx V4.1 software package.

Quantitative UHPLC-ESI-MS/MS analysis was performed using Agilent 1290 coupled with 6540 UHD Accurate-Mass Q-TOF spectrometer (Agilent Technologies) equipped with ESI source. Compounds were separated with the same Acquity UPLC HSS T3 column at 40°C. The mobile phases were used as follows: water with 10 mM ammonium formate and methanol with 10 mM ammonium formate as solvent A and B, respectively. Samples were loaded directly onto analytical column at the flow rate of 0.3 mL min$^{-1}$ of 100% solvent A. The injection volume was 2 µL or 8 µL for survey and MS/MS scans, respectively. Compounds were eluted from the column at the flow rate of 0.3 mL min$^{-1}$ with the following linear gradient: 0 min – 100% A, 3 min – 100% A, 6 min – 100% B, 9 min – 100% B and 12 min – 100% A. The eluted compounds were ionized in the negative ion mode with a capillary voltage of 2.5 kV in ESI source. Nitrogen was used as sheath and auxiliary gas (5 L/min, 150°C, 20 psi). Survey scans were recorded in the Q-TOF mass analyser at a resolving power of 20,000 in the $m/z$ range of 50 – 500. MS/MS spectra were recorded in cycle time 2 s. Collision energy values were optimized to obtain appropriate fragmentations of the studied compounds. Survey scans and MS/MS spectra were evaluated manually using Mass Hunter software (Agilent Technologies, Santa Clara, USA). Quantitative analysis performed for OSs, NOSs of 13BD SOA was based on a surrogate standard compound – sodium 1-pentyl sulfate (SP-OS). A stock solution was prepared in the concentration of 0.1 mg mL$^{-1}$. Tables S3 and S4 in SI presents mean peak area, recovery, LOD and LOQ levels obtained for pentyl sulfate. All calculations were performed for deprotonated ions (Figures S4).

**2.4 Chamber secondary HAPs, glyoxal, PAN, and APAN measurements**

Selected low molecular weight carbonyls and dicarbonyls (e.g., HAPs) were quantified by derivatization using 2,4-dinitrophenylhydrazine (Smith et al., 1989). Samples were collected at 0.5 L min$^{-1}$ for 20 min and derivatized in a 4 mL solution of acidified DNPH and then heated for 40 min at 70 ºC. Air samples were drawn through an impinger containing the

DNPH solution in acetonitrile. The resulting solutions were analyzed by high-performance liquid chromatography with a ultraviolet detector (HPLC/UV) (Smith et al., 1989). A 15-component hydrazone standard (comprising formaldehyde, acetaldehyde, acrolein, acetone, propionaldehyde, crotonaldehyde, methacrolein, butyraldehyde, 2-butanone, benzaldehyde, glyoxal, valeraldehyde, m-tolualdehyde, methylglyoxal, and hexaldehyde; AccuStandard, Inc.) at a free carbonyl concentration of 30 $\mu g$ cm$^{-3}$ for each component was used for calibration. Separate dihydrazone standards of glyoxal-DNPH and methylglyoxal-DNPH were also formulated. Carbonyls were separated using a Hewlett-Packard (HP) 1100 HPLC system having an Agilent Zorbax ODS 4.6 x 250 mm, 5-$\mu m$ column maintained at 30ºC eluted with binary acetonitrile-water gradient. A 10 $\mu L$ injection volume was used for all standards and samples. Carbonyls were quantified by UV absorption with a diode array detector set to 360 nm. Control and sample processing were managed with HP ChemStation software. Peroxyacetyl nitrate (PAN) and peroxyacryloyl nitrate (APAN) concentrations were determined using a gas chromatograph-pulse discharge electron capture detector (GC-PDECD; Model D-2, Valco Instrument Co., Houston, TX). The column made of fused-silica, had a 30-m x 0.53-mm ID with a 1.0-$\mu m$ Rtx-200MS liquid phase (Restek, Bellefonte, PA). It was operated at a carrier flow of 11.5 cm$^3$ min$^{-1}$. Samples were injected onto the head of the GC through a 1 cm$^3$ loop. Gas flows were adjusted for proper operation of the PDECD, that is, He at 30 cm$^3$ min$^{-1}$ with a 3 cm$^3$ min$^{-1}$ flow of a 5% CH$_4$/He which creates the standing current. The GC-PDECD system was operated at 60 ºC, and the total flow through the ECD was 44.5 cm$^3$ min$^{-1}$.

## 2.5. Aerosol acidity and liquid water content

Liquid water content (LWC) and the acidity of the aerosol aqueous phase generated during the smog chamber experiments were estimated using the Extended AIM Thermodynamic Model (E-AIM) (Wexler et al., 2002, Carlslaw et al., 1995, Clegg et al., 1992, 1994, 1995, 1998). The model was evaluated and used extensively for atmospheric applications (Pye at al., 2020). E-AIM does quantitative thermodynamic modeling of liquid/solid, liquid/gas and solid/gas equilibrium partitioning of aerosol components in an atmospheric system. We used the Model II Comprehensive variant of E-AIM, designed for systems containing H$^+$ - NH$_4^+$ - SO$_4^{2-}$ - NO$_3^-$ - H$_2$O. Nitrite ions were set to zero and no organic components were considered because of uncertain stoichiometry of dissociable hydrogen ions production during their formation and lacking reliable dissociation constants. Our input data included temperature, relative humidity, and formal molar composition of seed aerosol (NH$_4^+$, SO$_4^{2-}$, H$^+$).

## 3 Results and discussion

## 3.1 Chemical characterization

**3.1.1. Gas phase (GP) products.** The chamber conditions, including temperature and RH, are shown in Table 1. Table 1 also shows initial 13BD and NO concentrations, as well as reacted 13BD ($\Delta$HC), and steady state concentrations of O$_3$, and NOy (sum of NO, NO$_2$ and all oxidized odd-nitrogen species). Once the 13BD/NO mixture reached steady state under acidified or non-acidified conditions, NO had completely reacted while only a fraction of the initial 13BD was consumed. The reacted 13BD depended on RH and seed aerosol acidity (Table 1). Under these conditions, the steady state concentrations of NOy and O$_3$ given in Table 1 also depend on RH and seed aerosol acidity. O$_3$ and NOy produced during the reactions show a steady decrease with increasing RH (Figure 1a), except for NOy under non-acidified conditions, which did not change much across the RH used in this study. The results show that the increase in RH can reduce the maximum O$_3$ both under acidified and non-acidified seed aerosol (Figure 1a). Ozone was higher under non-acidified than acidified conditions across all RH used in this study. Figure 1b shows 13BD conversion efficiency (CE) (Liu et al., 2019b) as a function of RH for both experiments ER444 (acidified) and ER666 (non-acidified). Once the 13BD/NO mixture reached steady state, 13BD CE was found to be lower under acidified than non-acidified conditions, at ~8 % lower across all RH used in this study. A maximum of 80% and 74% CE was reached at lower RH (11–30%); and 74% and 68% at higher RH (49–60%) under non-acidified and acidified conditions, respectively (Table 1, Figure 1b). It has been previously reported that RH and the chemical make-up of the gas phase where the photooxidation occurs can influence the CE of 13BD (Wang et al., 2020). They found that 13BD photochemical reactivity first increased and then decreased with increased RH. Our study shows that the photochemical conversion of 13BD, either under acidified or non-acidified conditions, decreased with increasing RH, across all RH used in our study. The slight discrepancy at low RH among the present study and the findings from Wang et al (2020) is probably due to the experimental conditions used in both studies, including chamber mode (static vs. dynamic), chemical make-up of the

GP where oxidation occurs; RH (constant in our study vs only initial RH provided in Wang et al. experiments); and CE values taken after 24 hours reaction time vs. ~4 hours in our study. It has been reported that RH changes promote changes in OH radical concentrations (Hu et al., 2011). These changes may be one of the reasons for the RH affecting the photochemical conversion of 13BD in our study.

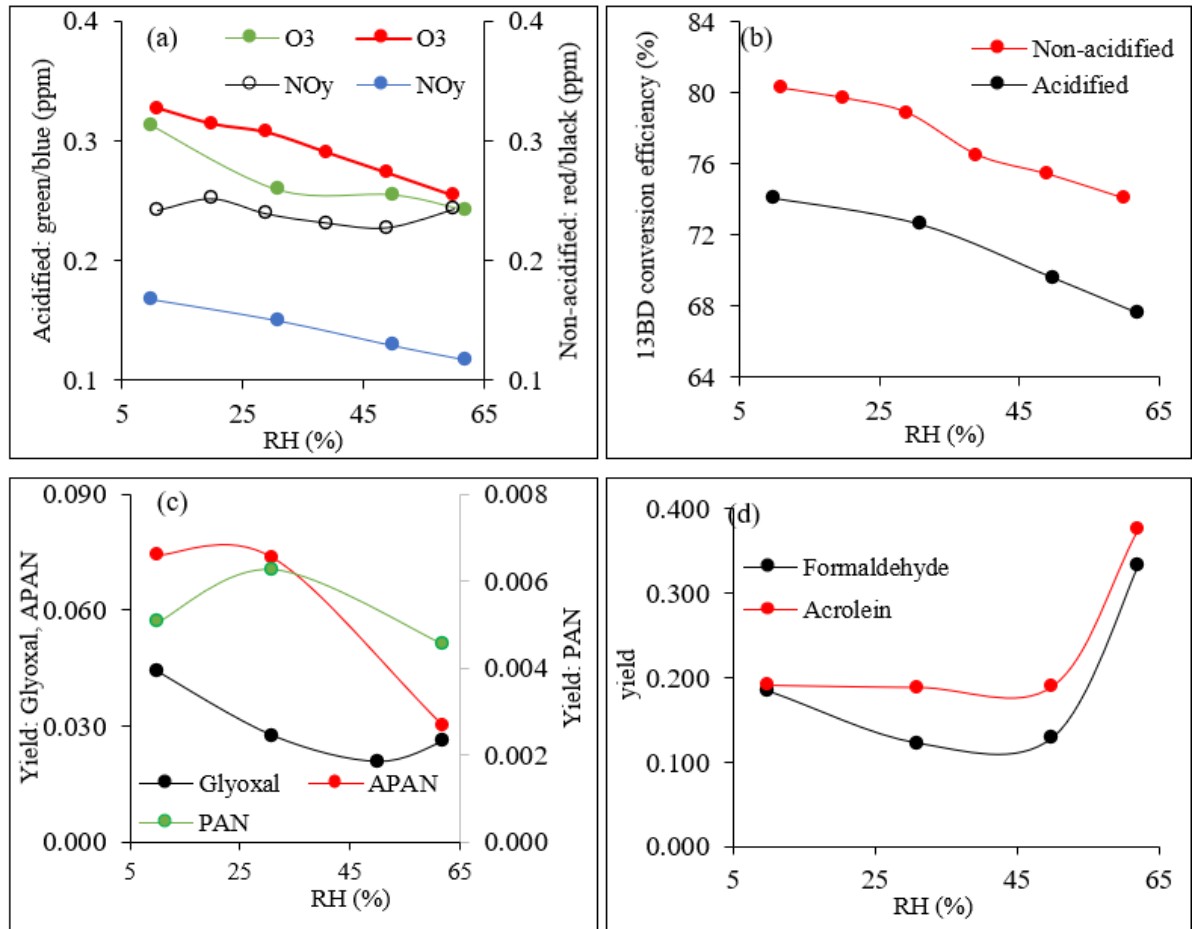

**Figure 1.** Influence of relative humidity on (a) $O_3$ and NOy concentrations (ppm) under acidified and non-acidified conditions; (b) 1,3BD conversion efficiency (%) under acidified and non-acidified conditions; (c) glyoxal, PAN, and APAN yield under acidified conditions; (d) formaldehyde and acrolein yields under acidified conditions. The smoothed lines in these plots were used solely as a visual guide.

Steady state concentrations and formation yields of formaldehyde, acrolein, glyoxal, PAN and APAN were measured only in experiment ER444 (Table 3), as the instruments were not working during experiment ER666. Under the conditions shown in Table 1 (ER444), high concentrations were observed for formaldehyde and acrolein, and to a lesser extent glyoxal, PAN and APAN (Table 3). This is the first time that experimental data have been presented for secondary HAPs from the oxidation of 13BD under a range of RH. Figures 1c-d show changes in the mass yield of these five species as a function of RH. The product mass yield was calculated as the product mass ($\mu$g m$^{-3}$) formed divided by the mass of reacted 13BD ($\mu$g m$^{-3}$). The reacted 13BD was calculated from the difference between the initial and steady-state concentrations. The concentrations of gas phase species were corrected for background and blank. Glyoxal and formaldehyde yields first exhibited a decrease, followed by a sharp increase as the RH increased. However, the yields of APAN and PAN first exhibited a slight increase,

followed by a decrease as the RH increased. Meanwhile, the yield of acrolein remained relatively steady as the RH increased, then increased sharply at 60% RH. This is consistent with previous findings for glyoxal formed from benzene and ethylbenzene

**Table 3.** Steady state gas phase concentrations (ppb) and formation mass yields (%) during 13BD photooxidation under acidified seed aerosol (ER444: (1/3 ammonium sulfate, 2/3 sulfuric acid by sulfate mass in precursor solution).

|  | Stage 1 | Stage 2 | Stage 3 | Stage 4 |
|---|---|---|---|---|
| RH (%) | 10 | 31 | 50 | 62 |
| Formaldehyde | 426.8 | 283.0 | 277.1 | 709.2 |
| Acrolein | 236.3 | 223.7 | 218.4 | 428.2 |
| Glyoxal | 52.9 | 32.9 | 23.0 | 28.6 |
| PAN | 2.9 | 3.6 | - | 2.4 |
| APAN | 38.8 | 38.5 | - | 14.5 |
| Formation yield (%) | | | | |
| Formaldehyde | 18.4 | 12.2 | 12.9 | 33.4 |
| Acrolein | 19.1 | 18.8 | 19.0 | 37.6 |
| Glyoxal | 4.4 | 2.7 | 2.1 | 2.6 |
| PAN | 0.5 | 0.6 | - | 0.5 |
| APAN | 7.4 | 7.4 | - | 3.0 |

(Jia and Xu, 2014). Glyoxal was found to partition into the particle phase and become hydrated at high RH and enhance the formation of SOA (Jia and Xu, 2014). The wall loss of particles is ~6% per hour. We expect gas phase products and 13BD wall to be negligeable as the experiments were conducted at steady state conditions, and in flow mode. The yield of formaldehyde and acrolein from the photooxidation of 13BD has been examined in a limited number of laboratory studies in the presence and absence of oxides of nitrogen. Berndt and Böge (2007) reported yields of formaldehyde (0.64 ± 0.08), and acrolein (0.98 ± 0.12) from OH radical reactions with 13BD conducted in a flow reactor. Kramp and Paulson (2000) reported the formation of acrolein from the ozonolysis of 13BD with a yield of (52 ± 7) %. In addition to primary emission, formaldehyde has been reported with high yields from the oxidation of most hydrocarbon photooxidation reactions, including 13BD (Bauwens et al., 2016, Wolfe et al., 2016). The formaldehyde yield reported in our study (Table 3) represents the overall yield (not only first-generation derived yield) considering all branches that can produce formaldehyde (second and further generation pathways); and consume formaldehyde (photolysis and reaction with OH radicals and other species, partitioning in the particle phase, etc.). Therefore, our yield is lower than those reported in the literature. Formaldehyde is used to estimate atmospheric VOCs emission rate (Bauwens et al., 2016; Cho et al., 2021 and reference therein). High emissions of 13BD in urban areas could be a significant source of secondary acrolein and formaldehyde and could influence urban atmospheric chemistry in these areas. PAN and APAN were also formed in this study with yield of ~ 0.5 for PAN and up to 7% for APAN at lower RH. A mass yield of 4% was measured for glyoxal at 10% RH and decreased as RH increased.

Insights from the present study can help elucidate the chemistry of other atmospherically relevant compounds, especially isoprene, the major non-methane hydrocarbon emitted globally (Guenther et al. 1995). While past work has identified reaction products from 13BD oxidation, we still do not understand explicit secondary HAPs formation and other oxygenated products, particularly over a range of seed aerosol acidity and RH. In addition, the uptake of some of these compounds (e.g., glyoxal) in the aerosol phase followed by heterogeneous sulfur chemistry can lead to SOA formation (Carlton et al., 2007; Liggio et al., 2005; Chan et al., 2010). The role of APAN in ambient SOA formation could be important, and might follow chemistry similar to MPAN, which only recently was found to play a role in SOA formation from isoprene (Tanimoto and Akimoto, 2001; Surratt et al., 2010).

### 3.1.2. Particle phase products.
In this study, up to six glass fiber (GF) filters were collected in each stage associated with ER444 and ER666 (Table 1). One filter from each stage was analyzed by UHPLC without derivatization. A second filter from each stage was analyzed by GC-MS following the derivatization procedure described above. The analysis at the molecular level of 13BD SOA generated from smog chamber experiments was based on the interpretation of GC-MS (in EI and CI modes) and UHPLC/ESI(-) HRMS mass spectra of derivatized and underivatized SOA products, respectively.

**GC-MS analysis.** The identification of reaction products from GC-MS analysis was based on comparisons between mass spectra of the derivatized samples and the authentic spectra in CI and/or EI mode, as well as on retention times, with the exception of compounds with no authentic standards. In this case, the identification was made based on the interpretation of fragment/adduct peaks in CI mode that permit determination of the number and identity of functional groups and the molecular weight of the derivative. A peak was associated with a reaction product only if its corresponding mass spectrum was consistent with the fragmentation pattern of the BSTFA derivatization reagent and was not detected in blank and background samples. In addition, an extensive evaluation was made by comparing products identified in this study with those identified in our previous work or other laboratories to determine the degree of consistency and to provide information on which chemical systems might be subject to the greatest uncertainties. Except for products bearing nitrate and sulfate groups, the BSTFA single derivatization technique provides good qualitative and quantitative analysis due to both its simplicity and efficiency (Jaoui et al. 2004). We previously reported the characteristic ions associated with BSTFA derivatization (Jaoui et al. 2014) paper for more information. In this GC-MS analysis, our main purpose was to compare and highlight the effect of relative humidity on the chemical characteristics of SOA products generated under acidified and non-acidified conditions. Additional analysis using LC-MS was carried out focusing mainly on products bearing sulfate and nitrate groups and reported in the LC-MS section below.

Figures 2, 3 and 4 present GC-MS extracted-ion chromatograms (EICs) associated with 13BD/NOx photooxidation experiments ER444 (Figures 2, 3) and ER666 (Figure 4), respectively. Each Figure contains EICs associated with each RH used in this study. According to obtained chromatograms, several compounds can be distinguished, i.e., 5 isomers of threonic acid, tartaric acid, threitol, meso-threitol, glyceric acid and erythrose. Tartronic acid and erythrose (MW 120) having both three OH active groups co-eluted under the GC-MS conditions used in this study, and their mass spectra were similar. The chromatograms in Figures 2, 3, and 4 can be directly compared because the chamber air sampled and the extraction and analysis were the same for each filter, although the relative abundance in the Y axis is different. Therefore, a relative contribution of products to SOA masses at various RH levels could be visually gained through these EICs. Two EICs were provided for the acidified experiment. They highlight oxygenated organic species (OOS) eluted early in the chromatograms (Figure 2), as well as several unknown (U1 through U7) products eluted later in the chromatograms (Figure 3). EICs of non-acidified smog chamber experiments highlight OOS eluted before 27 min and dimers eluted after 27 min (Figure 4). Compounds identified using GC-MS analysis are summarized in Table 4, which contains proposed structures for products identified, when possible, chemical formula, retention time, nomenclature, molecular weights of the derivatized ($MW_{BSTFA}$) and underivatized compounds (MW), as well as if the product is detected under acidified and/or non-acidified conditions. All of these products contain one or more of the flowing functional groups: alcoholic (-OH), acidified (-COOH) groups or both. We previously reported the formation of 13BD SOA products such as threitol, erythritol, glyceric acid, threonic acid and tartaric acid has been reported in field samples and chamber studies under low- and high-NOx conditions (Jaoui et al. 2014). Therefore, their identification is discussed only briefly in this study. Figure S5 presents mass spectra in CI mode of selected reaction monomer products observed in this study.

Two groups of unknown products were detected in both systems and absent in the background chromatograms. Although no structural information could be obtained, molecular weights of selected unknowns were tentatively obtained (Table 4). Their mass spectra were consistent with compounds bearing OH and COOH groups. Groups 1 and 2 consist of seven monomers (Figure 3, Table 4), and several dimers with mass spectra similar to those dimers (dimers 1-4) reported in Figure 3 (not shown in Table 4), respectively. Figure S6 shows mass spectra associated with these unknown dimers in CI mode. The formation mechanism for selected compounds under low-NOx conditions is possible given that the reactive uptake of 2,3-epoxy-1,4-butanediol (BEPOX) onto acidified aerosol seeds and under high-NOx conditions by further oxidation of acrylonyl peroxynitrate (APAN) to acrylic acid epoxide (AAE) (Jaoui et al., 2014). Additional organic compounds were present in the SOA collected that were not detected by the GC-MS derivative method, as can be seen from the presence of organonitrates and OSs using LC-MS. A detailed analysis is reported in the section below (LC-MS Analysis).

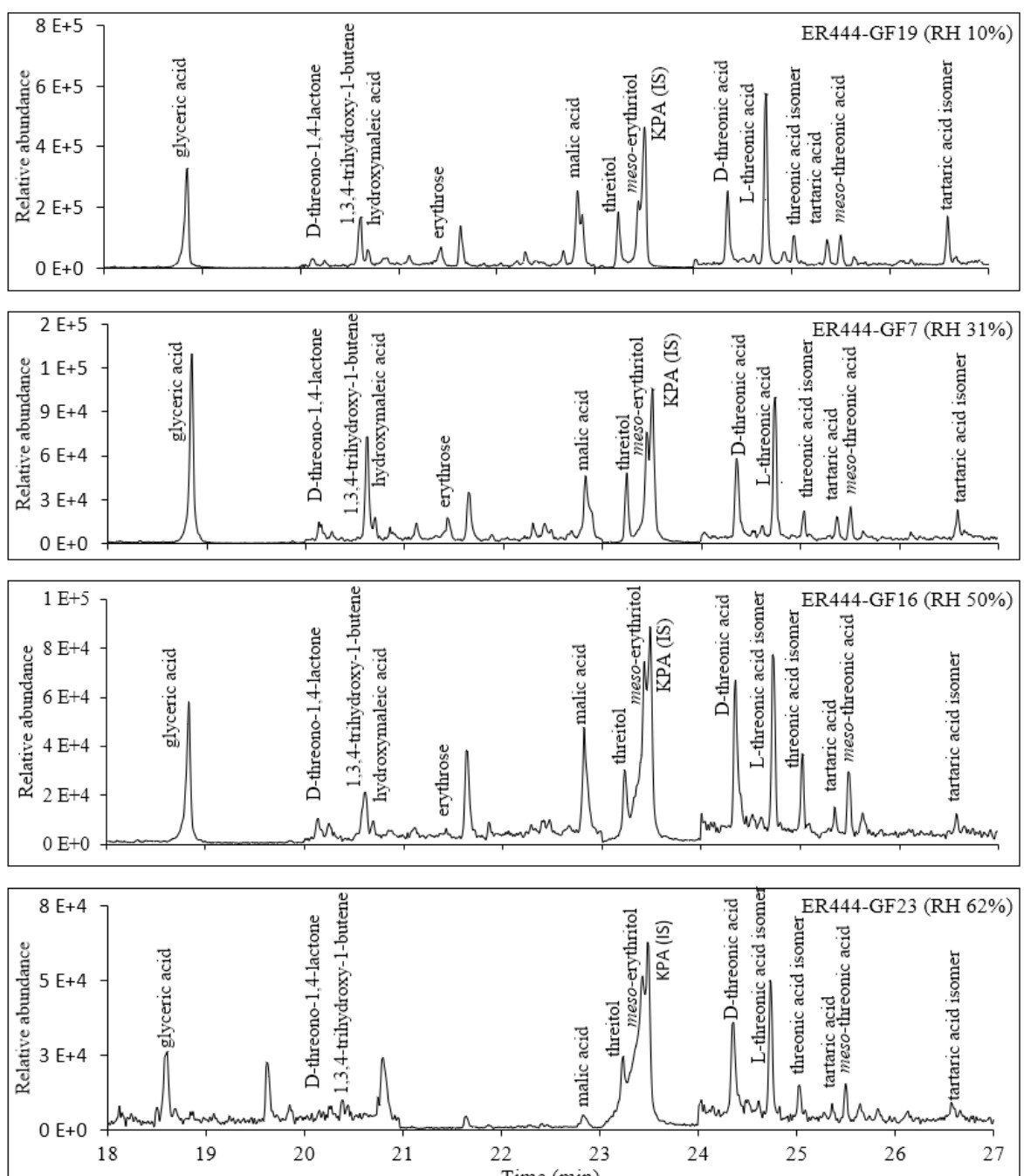

**Figure 2.** Extracted Ion Chromatograms (KPA: $m/z$ 165; ketopinic acid (IS)), ($m/z$ 307; glyceric acid), ($m/z$ 409; threonic acid, 5 isomers at 24.36, 24.53, 24.75, 25.04, and 25.51 min), ($m/z$ 321; erythrose), ($m/z$ 395; threitol, 2 isomers), ($m/z$ 247; D-threono-1,4-lactone), ($m/z$ 423; tartaric acid) for 1,3-butadiene/NOx photooxidation experiments ER444 (acidified seed) as a function of RH. Compounds detected as silylated derivatives. Intensity was multiplied by 5 between 20-23, and 24-27 min.

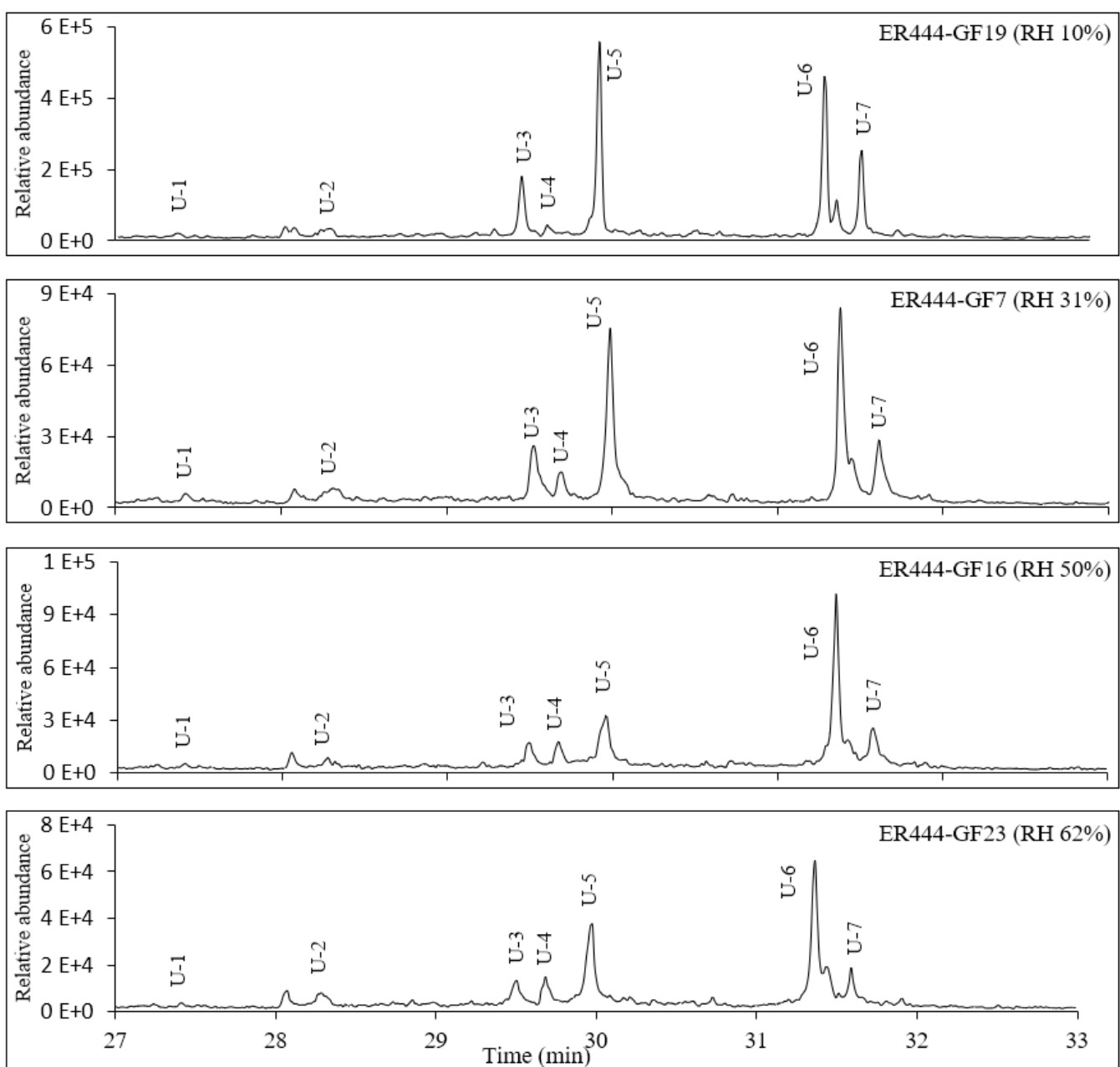

**Figure 3.** Portion of Extracted Ion Chromatograms at *m/z* 153, 229, 243, 259, 317, 405, and 481 for 1,3-butadiene/NOx photooxidation experiments ER444 under acidified seed aerosol as a function of RH (retention time between 27 and 32 min). Compounds detected as silylated derivatives. Only limited species are provided in this figure for clarity.

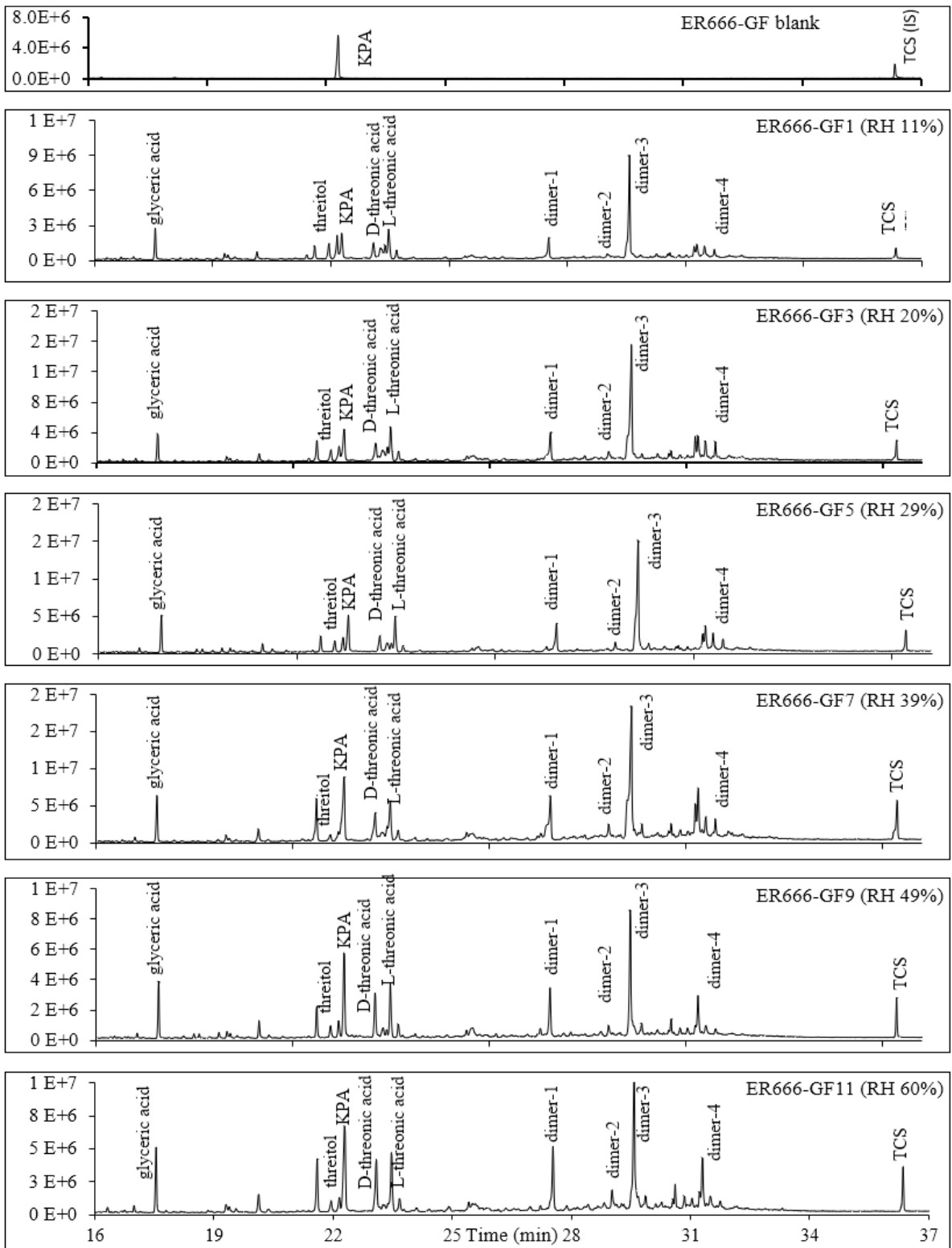

**Figure 4.** Extracted Ion Chromatograms showing monomeric compounds eluting before 25 min, and dimers eluting between 26 and 33 min for 1,3-butadiene/NOx photooxidation experiments ER666 as a function of RH. Dimer-1: oxalic acid-1,3,4-Trihydroxy-1-butene ester; dimer 2: oxalic acid-erythrulose ester; dimer 3: glyceric acid-1,3,4-trihydroxy-1-butene ester; dimer 4: hydroxymaleic acid-glycerol. Compounds detected as silylated derivatives. For clarity, not all identified compounds are included in Figure 3.

**Table 4.** Selected SOA reaction products identified from 13BD/NOx photooxidation under acidified and non-acidified conditions using GC-MS. Additional products are reported in Jaoui et al., 2014. Although dimers were detected under non-acidified conditions, they were more pronounced in the presence of acidified seed aerosol. Retention times (min) are those associated with ER666-GF7. MW: molecular weight of the underivatized compound; $MW_{BSTFA}$: molecular weight of the silylated derivatized compound. Unknown products are those where a formula has been obtained but the structure was not provided. Some organic acids were also identified using LC-MS.

| Chemical Formula (Rt: min) | *m/z* BSTFA Derivative (Methane-CI) | MW ($MW_{BSTFA}$) (g mol$^{-1}$) | Tentative Structure* | Nomenclature | Detection (acidified/non-acidified) |
|---|---|---|---|---|---|
| $C_3H_6O_4$ (17.54) | 189, 307, 205, 147, 323 | 106 (322) | | Glyceric acid (2,3-Dihydroxypropanoic acid) | both |
| $C_4H_6O_4$ (18.84) | 263, 247, 291, 73, 289 | 118 (262) | | D-Threono-1,4-lactone (3,4-Dihydroxydihydro furan-2(3$H$)-one) | both |
| $C_4H_6O_5$ (19.09 | 293, 321, 337, 365, 73 | 120 (336) | | Deoxythreonic acid | both |
| $C_4H_8O_3$ (19.30) | 305, 231, 73, 321, 349 | 104 (320) | | 1,3,4-Trihydroxy-1-butene | both |
| $C_4H_4O_5$ (19.38/19.55) | 231, 73, 203, 73, 305 | 132 (348) | | Hydroxy fumaric acid (2 isomers) | both |
| $C_4H_8O_4$ (20.11) | 321, 247, 203, 337, 231 | 120 (336) | | Erythrose | both |
| $C_4H_8O_4$ (20.12) | 247, 321, 337, 365, 377 | 120 (336) | | Tartronic acid | non-acidified |
| $C_4H_6O_5$ (21.59) | 335, 233, 73, 307, 351 | 134 (350) | | Malic acid | both |

| | | | | | |
|---|---|---|---|---|---|
| $C_4H_{10}O_4$ (21.95) | 395, 321, 205, 305, 411 | 122 (410) | | Threitol (1,2,3,4-Butanetetrol) | both |
| $C_4H_{10}O_4$ (22.15) | 395, 321, 205, 305, 411 | 122 (410) | | Erythritol (1,2,3,4-Butanetetrol) | both |
| $C_4H_8O_5$ (23.08) | 409, 217, 307, 73, 425 | 136 (424) | | D-Threonic acid (2,3,4-Trihydroxybutanoic acid) (2 isomers) | both |
| $C_4H_4O_5$ (23.18) | 333, 259, 349, 321, 73 | 132 (348) | | Hydroxymaleic acid | both |
| $C_4H_8O_5$ (23.47) | 409, 217, 307, 73, 425 | 136 (424) | | meso-Threonic acid (2R,3R)-2,3,4-trihydroxybutanoic acid (2 isomers) | both |
| $C_4H_4O_6$ (23.67/23.79) | 349, 275, 73, 185, 365 | 148 (364) | | Dihydroxymaleic acid (2 isomers) | both |
| $C_4H_6O_6$ (24.11) | 423, 305, 277, 321, 73 | 150 (438) | | Tartaric acid (2,3-dihydroxysuccinic acid) | both |
| $C_4H_6O_6$ (25.41) | 423, 393, 259, 191, 73 | 150 (438) | | Tartaric acid isomer (2,3-dihydroxysuccinic acid) | both |

**Oligomers**

| | | | | | |
|---|---|---|---|---|---|
| $C_7H_6O_{10}$ (27.54) | 303, 377, 421, 433, 465 | 176 (392) | | Oxalic acid-1,3,4-Trihydroxy-1-butene ester | both |
| $C_6H_8O_7$ (28.12/28.39) | 229, 319, 303, 257, 73 | 192 (408) | | Malic acid-glycolic acid ester (2 isomers) | both |
| $C_6H_{10}O_7$ (30.34/30.66) | 307, 467, 511, 365, 189 | 194 (482) | | Glyceric acid dimer ester (2 isomers) | non–acidified |

| Formula (RT) | Fragment ions | m/z (parent) | Structure | Name | Condition |
|---|---|---|---|---|---|
| $C_6H_8O_7$ (28.82/29.02) | 229, 319, 303, 257, 73 | 192 (408) |  | Oxalic acid-erythrulose ester (2 isomers) | both |
| $C_7H_{12}O_8$ (32.22, 33.03) | 207, 73, 569, 479, 405 | 224 (584) |  | Glyceric acid-threonic acid ester (2 isomers) | both |
| $C_7H_{12}O_8$ (31.38, 31.67, 31.88, 32.88) | 555, 207, 555, 570, 73 | 210 (570) |  | Glyceric acid-tetrol ester (4 isomers) | both |
| $C_6H_{12}O_6$ (25.93, 26.35) | 289, 363, 199, 407, 73 | 180 (468) |  | Glyceric acid-glycerol ester (2 isomers) | both |
| $C_7H_{12}O_6$ (29.33, 29.46, 29.51, 29.57, 29.60, 29.67, 29.84, 29.87, 30.56, 30.61) | 317, 391, 435, 447, 155 | 192 (480) |  | Glyceric acid-1,3,4-trihydroxy-1-butene ester (10 isomers) | both |
| $C_7H_{10}O_7$/ $C_8H_{14}O_6$ (30.84, 31.23, 31.30, 31.41, 31.49, 31.65, 31.73) | 405, 479, 389, 301, 73 | 206 (494) |  | Hydroxymaleic acid-glycerol ester/deoxythreonic acid-1,3,4-trihydroxy-1-butene ester (7 isomers) | both |
| $C_6H_8O_8$/ $C_7H_{12}O_7$ (30.37, 30.72) | 481, 407, 496, 241, 363 | 208 (496) |  | oxalic acid-threonic acid ester/Malic acid-glycerol ester (2 isomers) | both |

**Unknown reaction products**

| | Fragment ions | m/z (parent) | | | Condition |
|---|---|---|---|---|---|
| U-1 | 171, 243, 229 | (466) | - | - | acidified |
| U-2 | 259, 377, 333 | (332) | - | - | both |
| U-3 | 319, 229, 257 | (494) | - | - | acidified |
| U-4 | 257, 153, 185 | (332) | - | - | acidified |
| U-5 | 257, 153, 185 | - | - | - | both |
| U-6 | 243, 271, 333 | - | - | - | both |
| U-7 | 243, 271, 333 | - | - | - | both |

**LC-MS analysis.** In this section, we further explored the SOA analysis of smog chamber samples in order to confirm (1) the structures obtained for selected organic acids by GC-MS analysis (see organic acids section below), and (2) identification of OSs. The LC-MS analyses focused mainly on the identification of a variety of OSs and NOSs, as described below.

**Organic acids.** A set of the main organic acids including tartaric acid (TA), D-threonic acid (TrA), tartronic acid (TrtA) and malic acid (MA) were detected in the particle phase at all RH either under acidified, non-acidified conditions, or both using the LC-MS analysis. This is consistent with the results reported above using the derivatization technique followed by GC-MS analysis (Table 4, last column). The identification was based on comparing the results of smog chamber experiments ER444 and ER666 with standard compounds for these organic acids. Figure 5 shows EICs of TA, TrA, TrtA, and MA recorded for SOA samples (ER666) and authentic standards. The retention times and fragmentation patterns obtained for these SOA components are in good agreement with reference compounds. Recent studies suggest that TA and other carboxylic acids undergo heterogeneous OH reaction in aqueous solution (Cheng et al., 2016). Figures 6-9 show representative MS/MS spectra of TA, TrA, TrtA, and MA, along with proposed fragmentation patterns. Molecular formulas of the fragment ions were confirmed by mass measurements, but their structures were only tentative. Figure 6 top shows two high resolution product ion mass spectra of tartaric acid with characteristic fragment ions at $m/z$ 149 [$M_C$ – 1], 131 [$M_C$ - 18], 103 [$M_C$ - 46], 73 [$M_C$ - 76], and 59 [$M_C$ - 90] ($M_C$ is the molecular weight of the compound) recorded for both the smog-chamber sample and standard compound. The fragmentation pathway in Figure 6 shows that fragment ions with $m/z$ 131 and 87 are consistent with the loss of $H_2O$ and $H_2O$ and $CO_2$ from the deprotonated ion of TA ($m/z$ 149), respectively. Fragment ions with $m/z$ 105 and 73 are consistent with loss of $CO_2$ and $C_2H_4O_3$, respectively. Finally, fragment ions with $m/z$ 103 and 59 are consistent with the loss of ($H_2O$ + CO), and ($H_2O$ + $CO_2$ + CO), respectively. High resolution product ion mass spectra associated with TrA, TrtA, and MA are shown in Figures 7, 8, and 9, respectively.

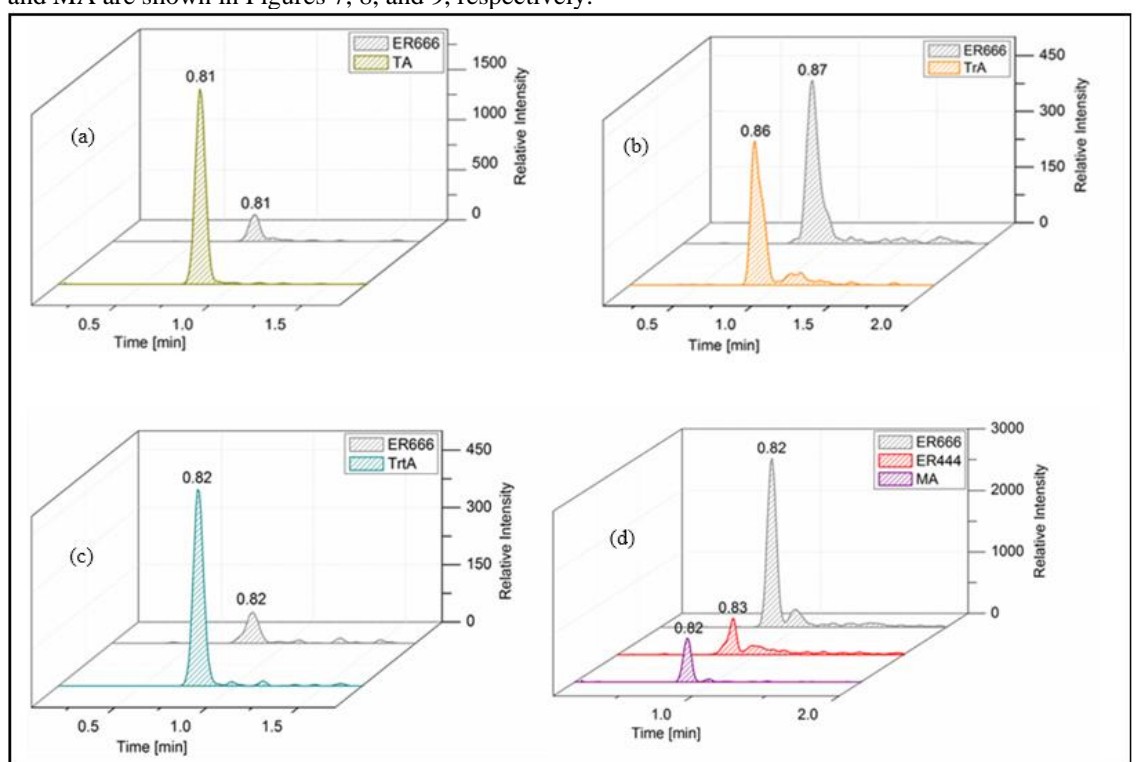

**Figure 5.** Extracted Ion Chromatograms (EICs) of: (a) tartaric acid (TA: MW 150); (b) threonic acid (TrA: MW 136); tartronic acid (TrtA: MW 120); and (d) malic acid (MA: MW 134) from smog chamber experiments and reference compounds.

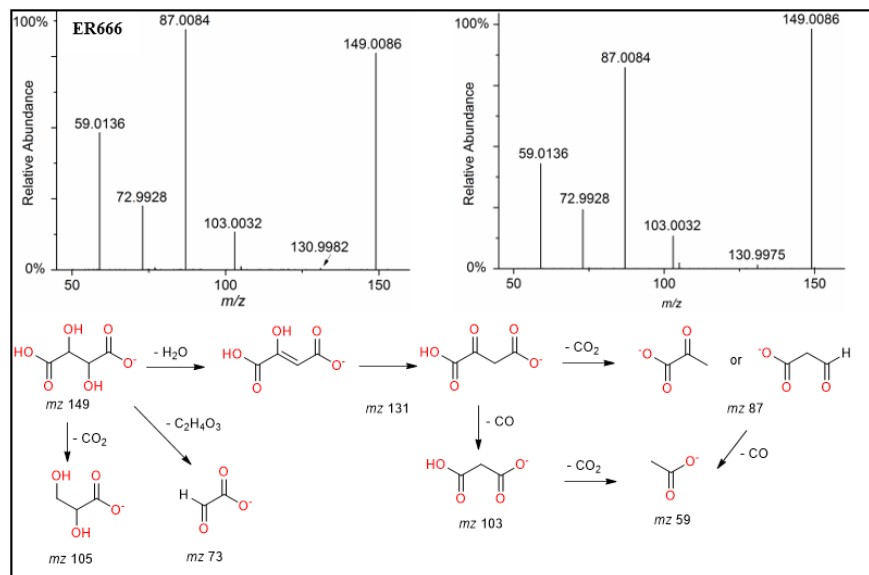

**Figure 6**. ESI(-) product ion mass spectra of tartaric acid (MW 150) eluting at RT = 0.81 min, registered for the ER666 non-acidified seed aerosol sample (left) and standard compound (right) along with a proposed fragmentation pathway.

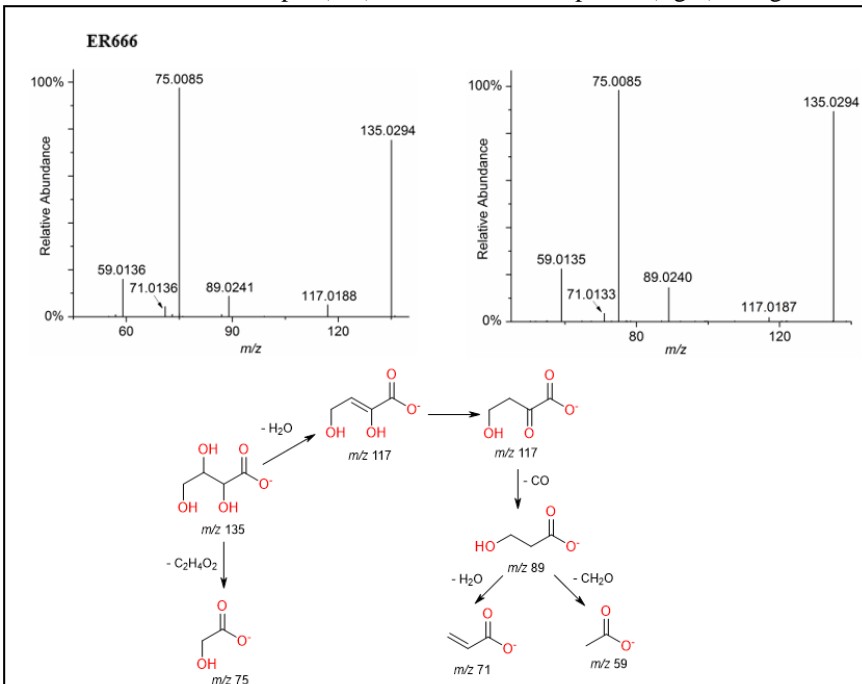

5 **Figure 7**. ESI(-) product ion mass spectra of threonic acid (MW 136) eluting at RT = 0.87 min, registered for a sample from the non-acidified ER666 experiments (left) and for the reference compound obtained by alkaline hydrolysis of D-threono-lactone (right) along with a proposed fragmentation pathway.

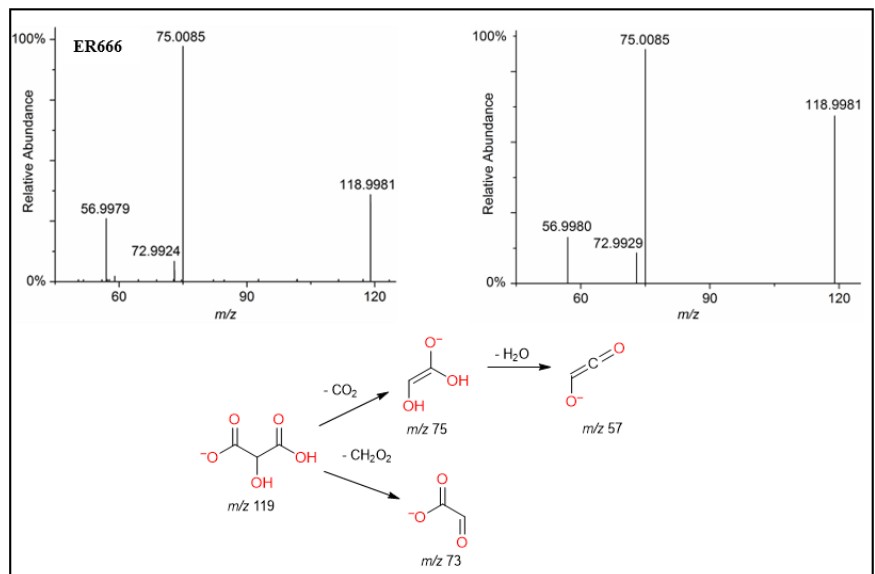

**Figure 8**. ESI(-) product ion mass spectra of tartronic acid (MW 120) eluting at RT = 0.82 min, registered for a sample from the non-acidified ER666 experiments (left) and the reference standard compound (right) along with a proposed fragmentation pathway.

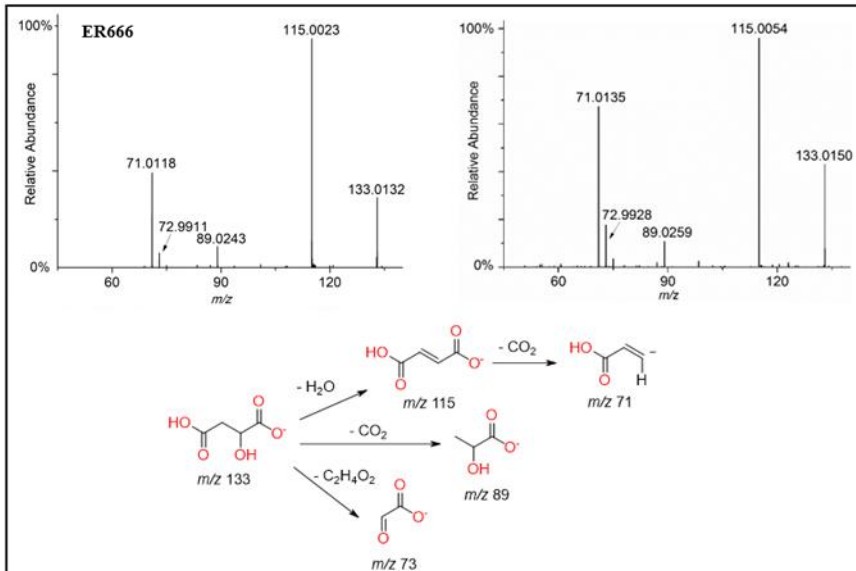

**Figure 9**. ESI(-) product ion mass spectra of malic acid (MW 134) eluting at RT = 0.82 min, registered for the ER666 non-acidified seed aerosol sample (left) and standard compound (right) along with a proposed fragmentation pathway.

**Organosulfates.** In this section, a tentative identification of OSs and NOSs is proposed based on accurate high-resolution mass data and MS/MS spectra, following the assumptions described in Nestorowicz et al. (2018). Briefly, this analysis was based on the deprotonated ions [M-H]⁻ and the corresponding fragmentation pathways. OSs were recognized by the loss of characteristic ions at $m/z$ 80 ($SO_3^-$), 96 ($SO_4^-$) and the presence of $m/z$ 97 ion ($HSO_4^-$) (Darer et al., 2011; Szmigielski 2016). The nitroxy-organosulfates and nitrosoxy-organosulfates were identified based on neutral loss of $m/z$ 63 ($HNO_3$) and $m/z$ 47 ($HNO_2$), respectively. All OSs identified have the same carbon backbone of butane, except glyceric acid-organosulfate (GA-

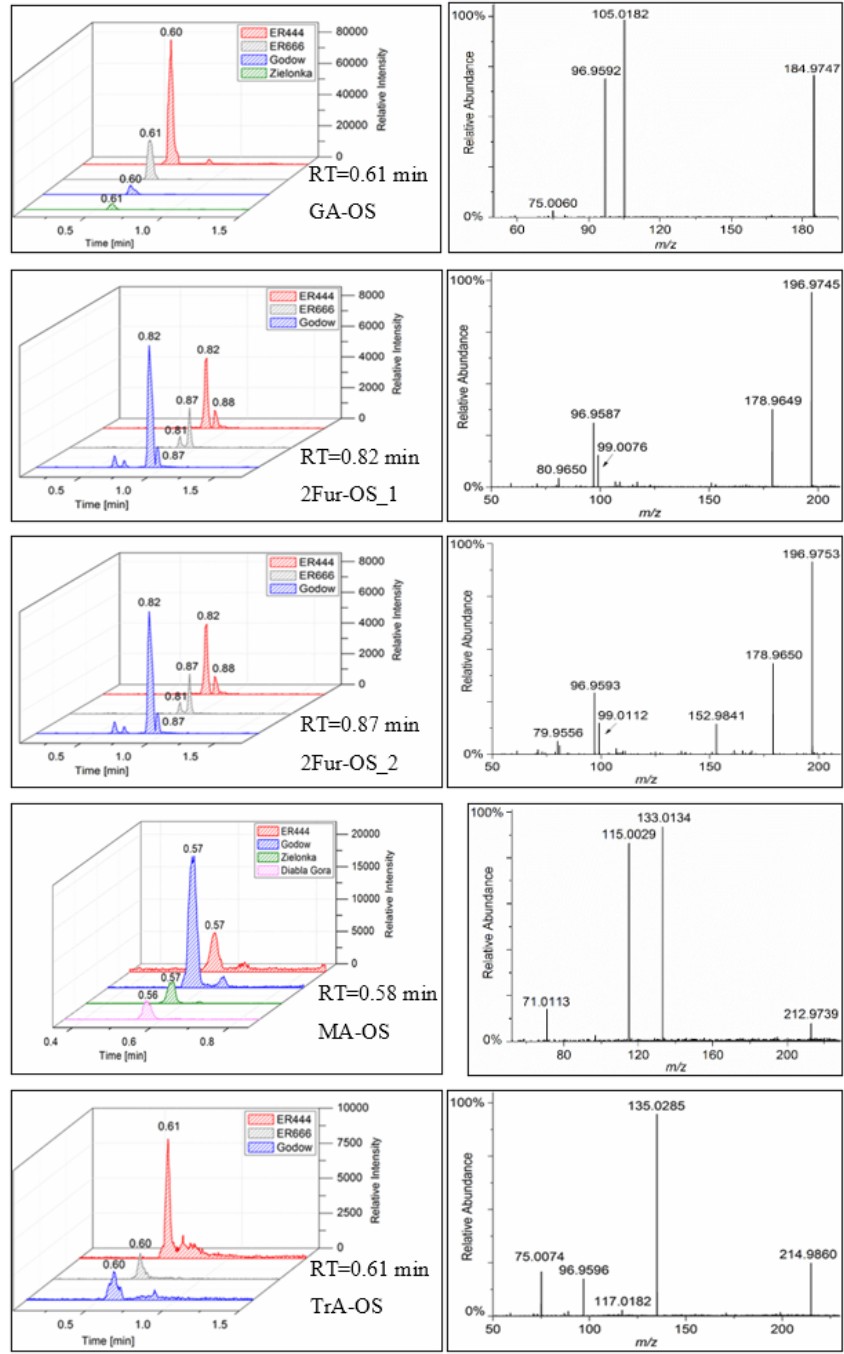

**Figure. 10.** Extracted Ion Chromatograms of selected organosulfates and their corresponding high resolution product ion mass spectra: MW 186 (glyceric acid-organosulfate: GA-OS); MW 198 (2(3*H*)-furanone, dihydro-3,4-dihydroxy-organosulfate: 2Fur-OS_1 and 2Fur-OS_2; 2 isomers); MW 214 (malic acid-organosulfate: MA-OS); and MW 216 (threonic acid-organosulfate: TrA-OS) obtained from smog chamber experiment ER444 and ambient samples (see section 3.3 for field samples analysis).

OS). The presence of $m/z$ 97 ion in a spectrum, corresponding to $HSO_4^-$, indicates that the hydrogen atom or a hydroxyl group is present at the carbon atom next to that bearing $HO–SO_2–O–$ moiety (Attygalle et al., 2001; Wach et al., 2020). For nitrosoxy- and nitroxy-OSs, the elimination of the $HNO_2$ and $HNO_3$ from the precursor ion was used for their identification. Figure 10

(left-side panels) shows EICs associated with SOA samples collected from ER444 and/or ER666 experiments and highlights selected newly identified OSs present also in ambient aerosol. Figure 10 (right-side panels) shows high-resolution product ion mass spectra associated with peaks of OSs shown in the corresponding left-side panels. Figure 11 shows proposed fragmentation patterns associated with the mass spectra shown in Figure 10. As an example, and for illustration purposes, Figure 10 top shows high resolution product ion mass spectrum of Ga-OS with characteristic fragment ions at $m/z$ 185 [MC –

1], 105 [MC - 80], 97 [MC - 88], and 75 [MC - 111]. The fragment ions detected at $m/z$ 97 and 105 are consistent with the diagnostic ion $HSO_4^-$, and the loss of $SO_3$ from the deprotonated GA-OS ion, respectively (Figure 11, top). For the remaining OSs shown in Figure 10, high resolution product ion mass spectra associated with 2Fur-OS (2 peaks), MA-OS, and TrA-OS are presented in Figure 11. Additional newly identified OSs including nitroxy- and nitrosoxy-OSs, detected solely in smog chamber experiments, are presented in SI, along with proposed fragmentation schemes (Figures S7–S12). Table 5 shows SOA

organic products bearing sulfate and nitrate groups detected from 13BD/NOx photooxidation under acidified and non-acidified conditions using UHPLC-ESI-MS/MS. Table 5 contains proposed structure for products identified, chemical formula, nomenclature, molecular weights of the compound, the main ions $m/z$, as well as if the product is detected under acidified and/or non-acidified conditions. Organosulfates were detected mainly under acidified conditions, however some of them were also detected under non-acidified conditions at low intensity, which may have originated from aerosol heterogenous reactions

involving ammonium sulfate present in the seed aerosol (Duporte et al., 2020).

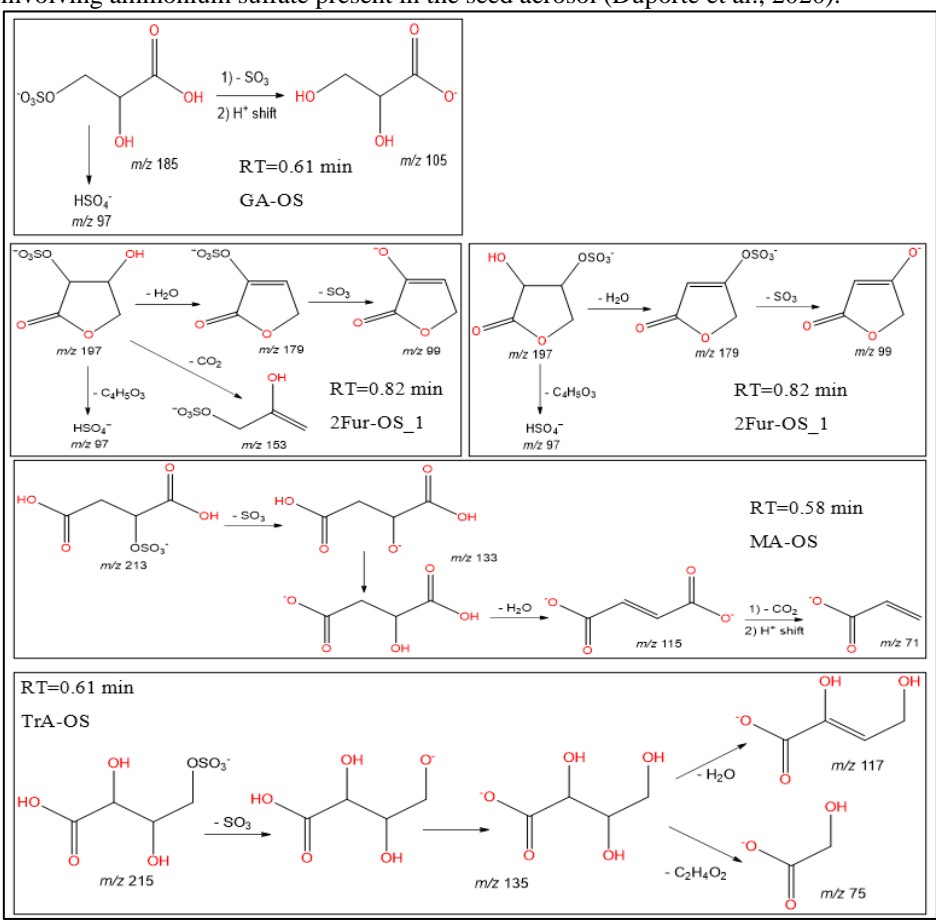

**Figure 11.** Proposed fragmentation pathways of organosulfates identified in Figure 10: MW 186 (glyceric acid-organosulfate: GA-OS); MW 198 (2(3*H*)-furanone, dihydro-3,4-dihydroxy-organosulfate: 2Fur-OS_1 and 2Fur-OS_2; 2 isomers); MW 214 (malic acid-organosulfate: MA-OS); and MW 216: threonic acid-organosulfate: TrA-OS).

**Table 5.** Selected SOA reaction products bearing sulfate and nitrate groups detected from 13BD/NOx photooxidation under acidified and non-acidified conditions using UHPLC-ESI-MS/MS.

| Chemical Formula | *m/z* Main Ions | MW (g mol$^{-1}$) | Tentative Structure * | Nomenclature | Detection (acidified and/or non-acidified conditions) |
|---|---|---|---|---|---|
| $C_4H_8O_6S$ | 183, 153, 97 | 184 |  | 2-butanone, 1,4-dihydroxy-organosulfate | both |
| $C_3H_6O_7S$ | 185, 105, 97 | 186 |  | glyceric acid-organosulfate | both |
| $C_4H_6O_7S$ | 197, 179, 153, 99, 97 | 198 |  | 2(3*H*)-furanone, dihydro-3,4-dihydroxy-organosulfate | acidified |
| $C_4H_8O_7S$ | 199, 181, 169, 139, 97 | 200 |  | 2-butanone, 1,3,4-trihydroxy-organosulfate | acidified |
| $C_4H_{10}O_7S$ | 201, 97 | 202 |  | 1,2,3,4-butanetetrol-organosulfate | both |
| $C_4H_{10}O_7S$ | 213, 133, 115, 71 | 214 |  | malic acid-organosulfate | both |
| $C_4H_8O_8S$ | 215, 135, 117, 97, 75 | 216 |  | threonic acid-organosulfate | both |
| $C_4H_9O_8SN$ | 230, 212, 194, 183, 169, 165, 138, 97 | 231 |  | 1,2,3,4-butanetetrol-nitrosoxy-organosulfate | acidified |
| $C_4H_9O_9SN$ | 246, 183, 170, 152, 139, 97, 96 | 247 |  | 1,2,3,4-butanetetrol-nitroxy-organosulfate | both |
| $C_4H_7O_{10}SN$ | 260, 183, 169, 139, 97 | 261 |  | threonic acid-nitroxy-organosulfate | both |

## 3.2. Quantitative/semi-quantitative analysis
### 3.2.1. Effect of relative humidity on SOA products

**GCM-S analysis.** The concentrations of selected reaction products reported in table 4 were measured. The quantitative analysis was conducted using d-threitol as surrogate standard and cis-ketopinic acid (KPA) as internal standard, except for malic acid and tartaric acid, because authentic standards were not commercially available. Although calibration curves were obtained for

several surrogates (Tables S1, S2, Figure S1), d-threitol was used because its mass spectrum shares several fragment ions with the majority of reaction products identified with GC-MS (Table 4). Although quantitative information is very important for source apportionment and health effects purposes, due to the non-availability of commercial authentic standards in this study, our main objective was to gain insight into the identification and distribution of 13BD SOA products and their relative contribution as a function of RH and seed aerosol acidity. Therefore, the quantitative/semi-quantitative analysis either with GC-MS or LC-MS should be regarded as such. Figures 12 and 13 display a semi-quantitative analysis of selected reaction products identified under acidified (ER444) and non-acidified (ER666) conditions across the RH used in this study (Table 4). Figure 12 shows estimated concentrations of fourteen monomers (top), four dimers (middle), and seven unknowns (bottom) identified in ER444 as a function of RH. The total SOC mass concentration during ER444 experiment ranged from 60.3 at RH = 10% to 31.1 ug/m$^3$ at RH 62%. The concentrations of individual (1) monomers ranged between below detection limit to about 1064 ng/m$^3$; (2) dimers between 16 and 497 ng/m$^3$; and (3) unknowns between 18 and 555 ng/m$^3$. These values are typical of the range of values often seen for individual compounds detected in SOA chamber samples from the oxidation of other hydrocarbons. The concentration of the majority of products decreased with increasing RH with a few exceptions. The major contributors to 13BD SOA across all RH were glyceric acid reaching up to 1064 ng/m$^3$ at RH 31%, followed by glyceric acid-1,3,4-trihydroxy-1-butene ester dimer (second major component detected) reaching an estimated value of 497 ng/m$^3$ at 31% RH. The three major monomers observed after glyceric acid were threitol (all isomers), threonic acid (all isomers), and 1,3,4-trihydroxy-1-butene (Figure 12: top). All dimers and unknowns (U1-U7) were detected at a relatively high concentration (Figure 12 middle, bottom). Similar to aerosol production and monomers, they were found to be highly sensitive to the initial conditions, in particular RH.

Figure 13 shows estimated concentrations of fourteen monomer (top), four dimer (middle), and four unknown (bottom) products identified in ER666 as a function of RH using BSTFA derivatization. The total SOC mass concentration during ER666 experiment ranged from 45.1 ug/m$^3$ at RH = 11% to 24.7 ug/m$^3$ at RH 60%, values lower than those measured under acidified seed aerosol. The concentrations of individual (1) monomers ranged between below detection limit to about 207 ng/m$^3$; (2) dimers between 10 and 178 ng/m$^3$ except for glyceric acid-1,3,4-trihydroxy-1-butene ester which was observed at very high concentrations between 142 (RH = 60%) and 798 (RH = 11%) ng/m$^3$; and (3) unknowns between 18 and 555 ng/m$^3$. Note, only U3, U6 and U7 were observed under non-acidified conditions (Figure 13 bottom) and at relatively low concentrations compared to those under acidified conditions. Again, these values are typical of the range of values often seen for individual compounds detected in SOA chamber samples from the oxidation of other hydrocarbons. The concentration of the majority of products decreased with the increase of RH. Threonic acid isomers, threitol isomers, followed by glyceric acid were among the highest species observed under non-acidified conditions, respectively. For example, the concentrations of *meso*-threonic acid ranged between 82 ng/m$^3$ at 60% RH, and 207 ng/m$^3$ at 11% RH, and increased with decreasing RH. The formation of individual compounds as under acidified conditions were found to be highly sensitive to the initial conditions, in particular RH.

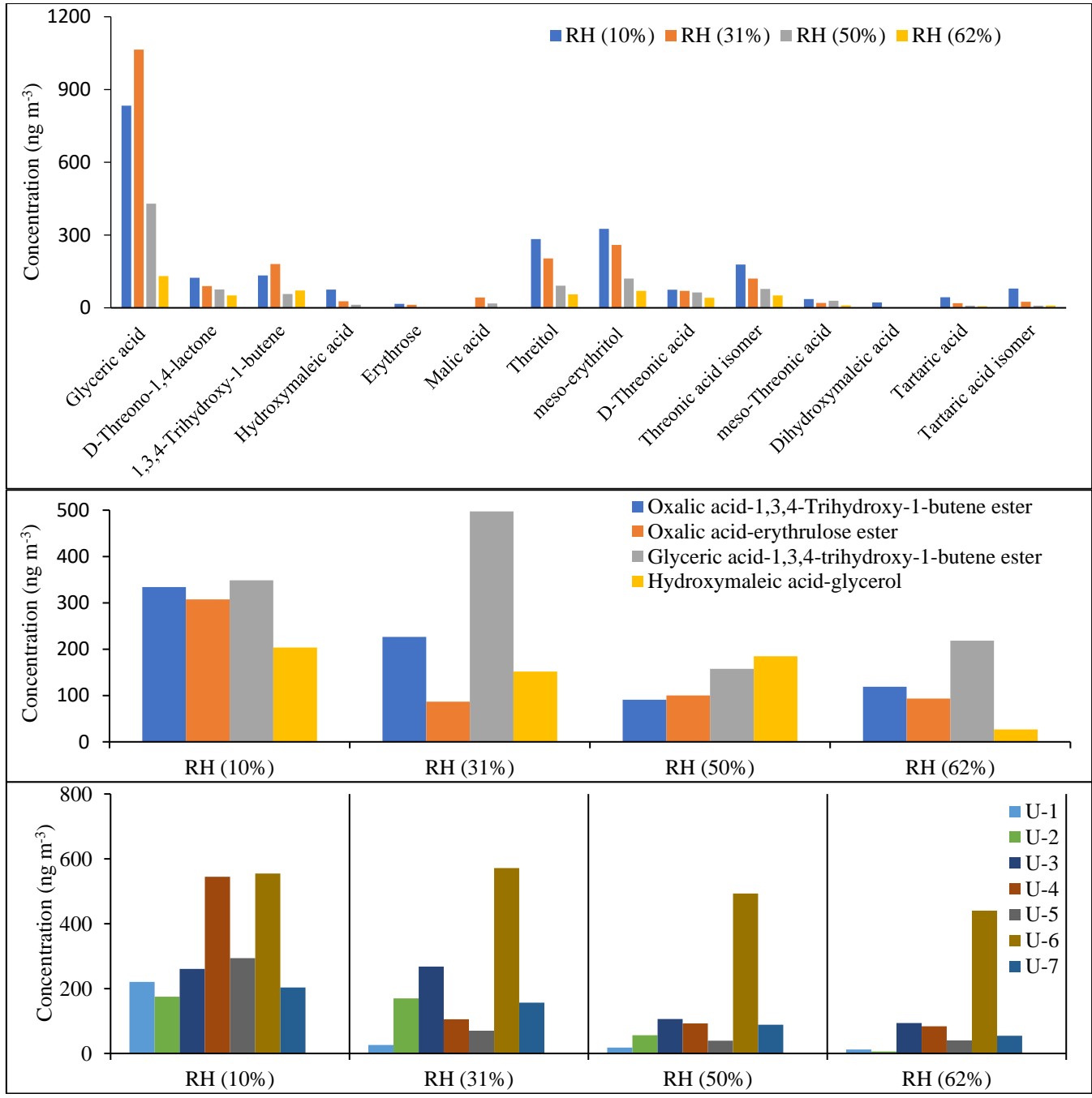

**Figure 12**. Estimated concentrations of selected SOA products as a function of RH obtained from ER444 experiment using BSTFA derivatization. d-Threitol was used as external surrogate standard for the semi-quantitative analysis for all products, except d-threitol, tartaric acid, and malic acid for which authentic standards were used. Cis-ketopinic acid was used as internal standard for all reaction products. See Table 3 and its description for additional information about the identity of these products.

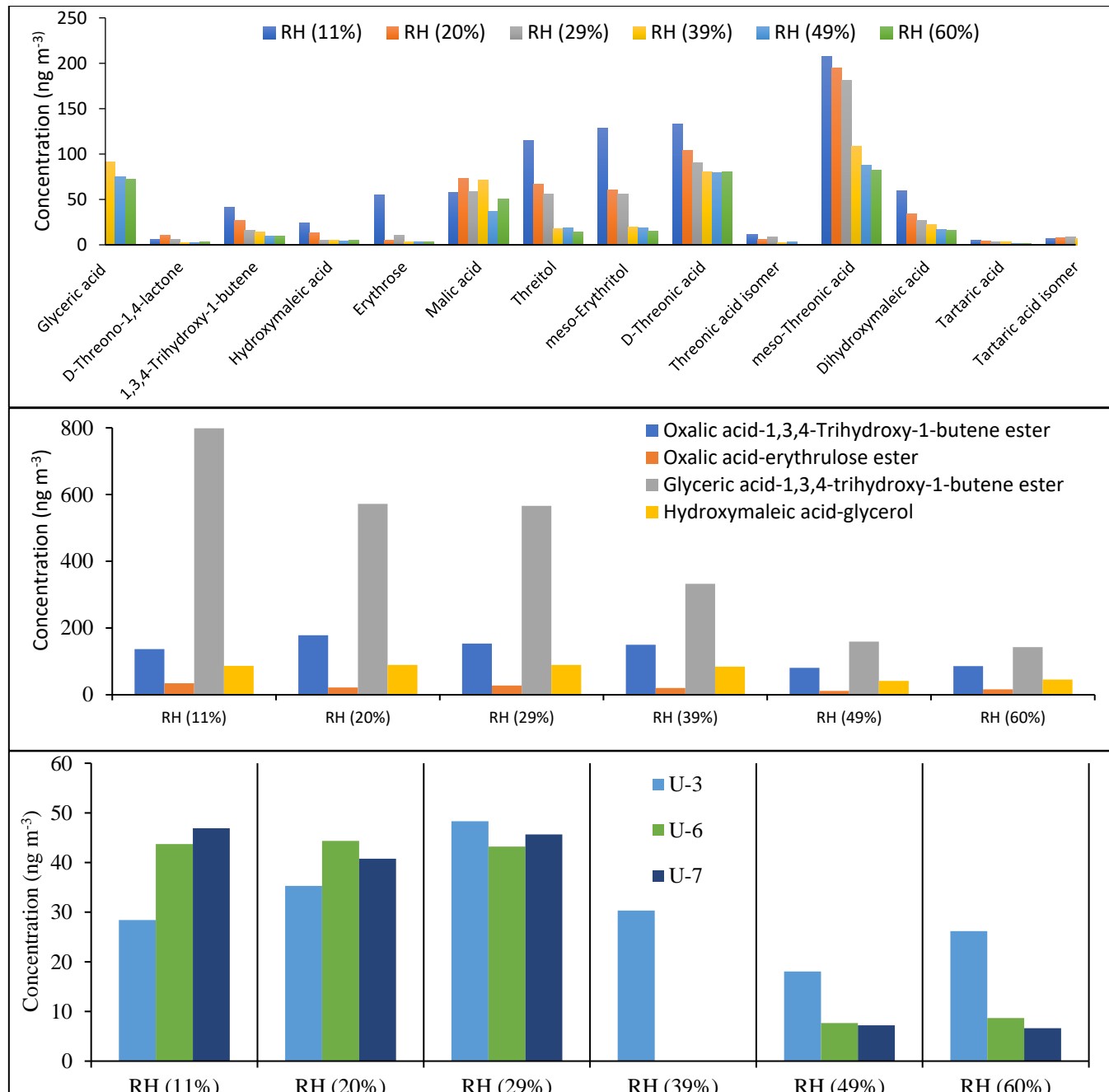

**Figure 13**. Estimated concentrations of selected SOA products as a function of RH obtained from ER666 experiment using BSTFA derivatization. d-Threitol was used as external surrogate standard for the semi-quantitative analysis for all products, except d-threitol, tartaric acid, and malic acid for which authentic standards were used. Cis-ketopinic acid was used as internal standard for all reaction products. See Table 3 and its description for additional information about the identity of these products.

**LC-MS analysis.** The concentration of organosulfates shown in table 5 were measured using LC-MS analysis, except for $C_4H_7O_{10}SN$ (*m/z* 261). The concentrations were obtained using sodium 1-pentyl sulfate (SP-OS) as a surrogate, because

authentic standards of 13BD OS are not commercially available. Recovery, limit of detection (LOD), and limit of quantification (LOQ) are reported for SP-OS in Table S4. Figure S4 displays calibration curves used for quantitative analysis associated with SP-OS. Figure 14 displays the concentrations measured using SP-OS as a surrogate for these OS under acidified (top) and non-acidified (bottom) conditions as a function of RH. High concentrations across all RH were measured for 1,2,3,4-butaneterol-NOS (*m/z* 246); and at relatively moderate concentrations for 1,2,3,4-butanetetrol-OS (*m/z* 201); glyceric acid-OS (*m/z* 185)

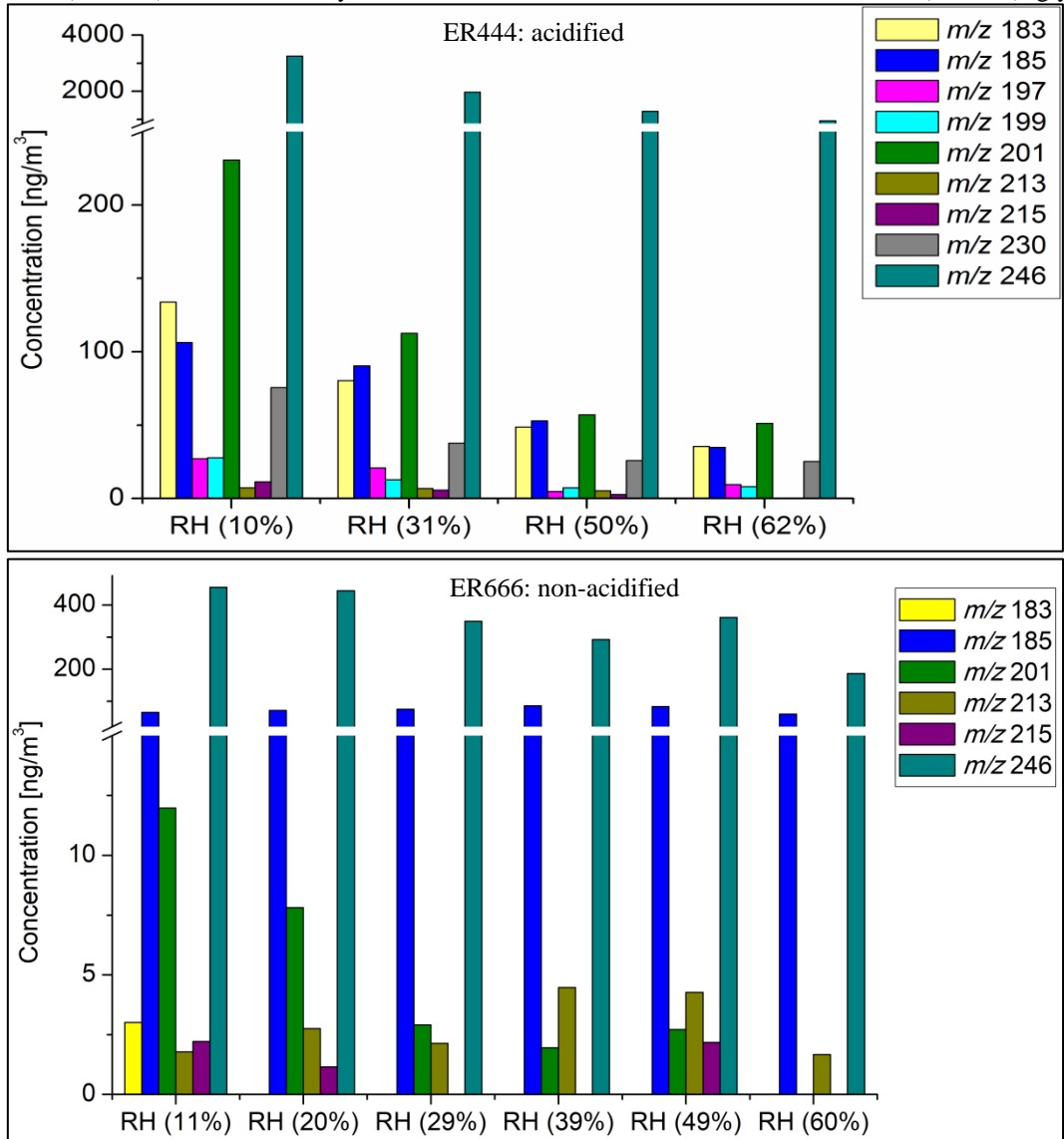

**Figure 14**. Estimated LC-MS concentrations (ng m$^{-3}$) of OS and NOS reaction products detected in SOA (Table 5) under acidified (ER444: top), and non-acidified (ER666: bottom) as a function of RH. The chemical name associated with *m/z* are provided in Table 5.

and 2-butanone,1,4-dihydroxy-OS (*m/z* 183). The highest values were observed under acidified seed experiment (ER444) at low RH and decreased with increasing RH, exceeding 3000 ng/m$^3$ at 30% RH (Figure 14). Similar mechanistic pathways can lead to the formation of these OS through acid-catalyzed multiphase chemistry of BEPOX (e.g., for 1,2,3,4-butaneterol-OS,

*m/z* 201; 1,2,3,4-butanetetrol-NOS, *m/z* 246) and AAE (e.g., for glyceric acid-OS, *m/z* 185) as those proposed for isoprene (Nestorowicz et al., 2018). In general, the increase of RH in the chamber resulted in decreased concentrations of OSs, although some OSs observed were not RH dependent (e.g., malic acid-OS: *m/z* 213, glyceric acid-OS: *m/z* 185). These results are consistent with previously reported data for isoprene SOA components (Nestorowicz et al. 2018). 2-Butanone, 1,4-dihydroxy-
OS (*m/z* 183), (3*H*)-furanone, dihydro-3,4-dihydroxy-OS (*m/z* 197), 2-butanone, 1,3,4-trihydroxy- OS (*m/z* 199) and 1,2,3,4-butanetetrol nitrosoxy-organosulfate (*m/z* 230) and are found to be considerably abundant under acidified conditions. Glyceric acid-OS was found to be RH dependent under acidified conditions only, results consistent with methyl-glyceric acid-OS reported in isoprene SOA (Nestorowicz et al., 2018). 1,2,3,4-butanetetrol-OS (*m/z* 201) was detected under both conditions and RH dependent, though with high abundance under acidified conditions. Malic acid-OS (*m/z* 213) and threonic acid-OS
(*m/z* 215) were detected in trace amounts under both conditions, acidified and non-acidified. Threonic acid-NOS (*m/z* 260) was detected also under both conditions in trace amounts, however at a level below LOD (not shown in Figure 14). The most abundant compound produced under acidified and non-acidified conditions was 1,2,3,4-butanetetrol-NOS (*m/z* 246). The estimated concentrations were 10-fold higher than those obtained for other 13BD SOA components. The amounts at the lowest RH reached ~3200 ng/m$^3$ and ~450 ng m$^{-3}$ under acidified and non-acidified seed conditions, respectively. The estimated
concentrations of detected 13BD SOA components established by LC-MS and GC-MS techniques are presented in SI (Tables S5-S8).

### 3.2.2. Effect of seed aerosol acidity on 13BD SOA products

The TIC chromatograms shown in Figure 15 can be compared because: (1) the sampling time, flow rate, and analytical
processes were the same for each experiment, (2) they were originated from two stages of experiments ER444 (blue: acidified) seed) and ER666 (red: non-acidified seed) conducted under similar RH (50%) though at slightly different temperature (Table 1), and (3) delta-13BD was the same for each stage. Figure 15 displays parts of the TICs associated with SOA originated from the photooxidation of 13BD at similar relative humidity (~50%) under acidified (blue) and non-acidified (red) seed aerosol for the portions between 17-27 min (top), and 27-33 min (bottom). A detailed examination of mass spectra associated with each
peak detected in these chromatograms reveals significant changes in the composition of particles depending on the acidity of the original seed aerosol. As shown in Table 4 (last column), the majority of compounds were detected in both chromatograms, except in the blank, although at variable abundance. The presence of sulfuric acid in the seed aerosol (ER444) resulted in a relatively substantial (1) increase in the intensity of the majority of peaks detected after 27 min, and (2) decrease in the intensity of the majority of peaks detected below 27 min. Compounds eluted before 27 min were more abundant under acidified
conditions (right Y axis: red) than under non-acidified conditions (left Y axis: blue). This is consistent with the occurrence of chemical reactions probably in the particle phase in the presence of acidified seed aerosol. The changes in seed aerosol acidity content between ER444 and ER666 resulted in significant decreases in product concentrations associated with monomers as shown in Figures 12 and 13 (top panels: e.g., threitols, threonic acid); and an increase in intensity as well as formations of dimers (Figures 12, 13: middle panels) and organosulfates (Figure 14). It is likely that the production of dimers and
organosulfates is partially suppressed in the absence of acidified seed aerosol in the system. The presence of acidified seed aerosol also resulted in a pronounced increase in the concentration of unknown compounds (Figures 12, 13: bottom panels).

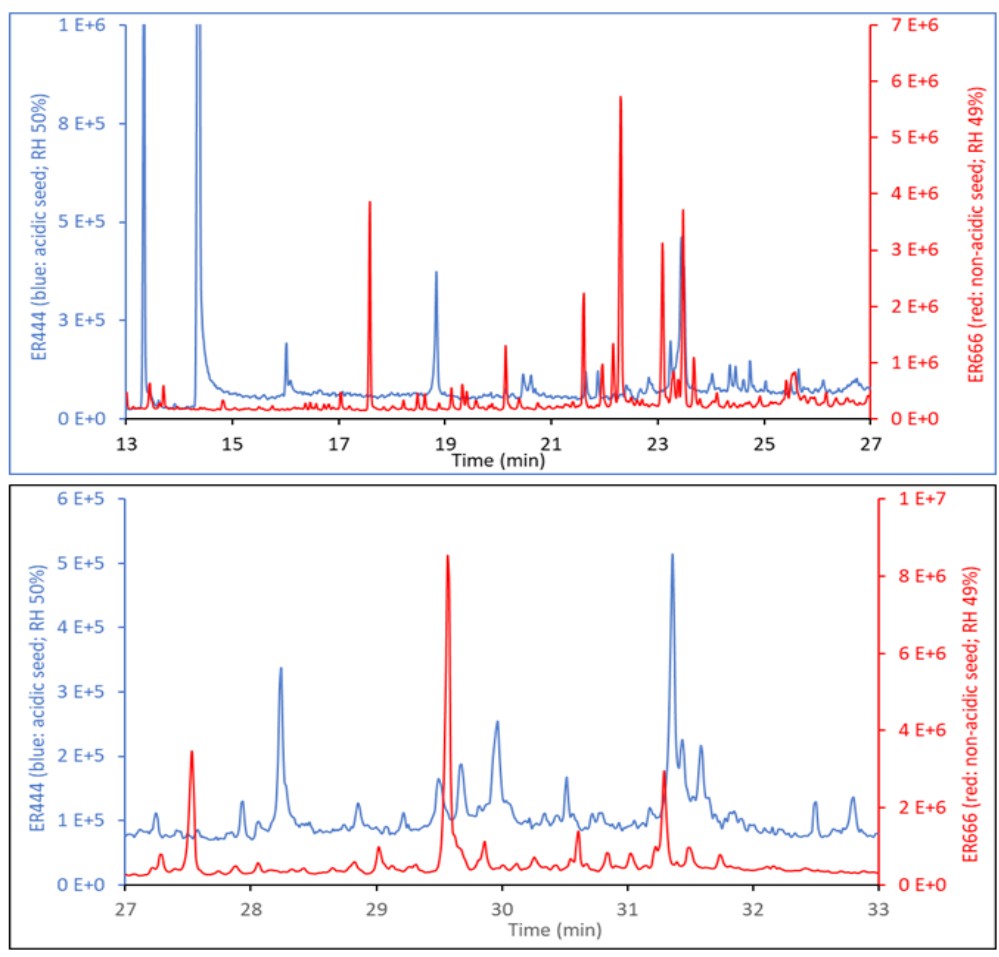

**Figure 15.** Parts of the Total Ion Chromatograms associated with SOA originated from the photooxidation of 13BD at similar relative humidity (~50%) under acidified (blue) and non-acidified (red) seed aerosol for portion between 17-27 min (top), and 27-33 min (bottom).

### 3.2.3 Relation of humidity and acidity of particle aqueous phase.

The acidity of the particle aqueous phase was estimated using the E-AIM model with input data given in the SI (section S2). The aqueous phase in ER666 experiment (non-acidified seed) was absent across the whole RH range, therefore only organic phase was present. In ER444 experiment (acidified seed), the particle aqueous phase was always present. Table 6 shows $H^+$ activity coefficient, the activity ($a_{H+}$), and the LWC associated with the particle aqueous phase across the whole RH range in ER444. The acidity of the aqueous phase decreased and LWC increased with increasing RH. The pH ($a_{H+}$) of the aqueous phase increased from -1.79 at 10% RH to ~0.40 at 62% RH. The contribution of organic acids to the formation of the particle aqueous phase and its acidity was not considered. It could be only marginal in acidified-seed experiments, where the highest possible amount of dissociable hydrogen ions from organic acids was less than 3% of the amount contributed by the sulfuric acid used to prepare seed aerosol. In the non-acidified seed experiments, organic acids could have more profound influence, so the results of our modeling should be taken as a first approximation and verified as soon as the precise stoichiometry of the acids formation and reliable dissociation constants of those acids are available.

**Table 6.** Water contents, pH and acidity of the particle aqueous phase in ER444 smog-chamber experiment.

| RH | T | $[H^+]_{aq}$ | $H^+$ activity coefficient | $H_2O$ | $[H^+]$ | pH ($[H^+]$) | $a_{H+}$ | pH ($a_{H+}$) | LWC |
|---|---|---|---|---|---|---|---|---|---|
| (%) | (K) | (mol m$^{-3}$) (air) | | (cm$^3$ m$^{-3}$) (air) | (mol dm$^{-3}$) (solution) | | activity | | (m$^3$ m$^{-3}$) |
| 10 | 295.55 | $1.31800\times10^{-7}$ | $3.39800\times10^2$ | $1.54541\times10^{-5}$ | 8.52848 | -0.931 | $6.15\times10^1$ | -1.789 | $1.54541\times10^{-11}$ |
| 31 | 295.75 | $1.65180\times10^{-7}$ | $6.71600\times10^1$ | $3.60989\times10^{-5}$ | 4.57576 | -0.660 | 6.69 | -0.825 | $3.60989\times10^{-11}$ |
| 50 | 295.35 | $1.89790\times10^{-7}$ | $1.42800\times10^1$ | $4.88070\times10^{-5}$ | 3.88858 | -0.590 | 1.13 | -0.052 | $4.88070\times10^{-11}$ |
| 62 | 295.35 | $2.08810\times10^{-7}$ | 5.83300 | $5.96244\times10^{-5}$ | 3.50209 | -0.544 | $3.99\times10^{-1}$ | 0.399 | $5.96244\times10^{-11}$ |

The SOC concentration and carbon yield (Table 1) increased with decreasing RH under acidified and non-acidified conditions. The carbon yield was higher under acidified conditions than non-acidified conditions and increased with increasing acidity of the aerosol aqueous phase. On the contrary, the 13BD conversion efficiency was higher at non-acidified conditions than acidified (Figure 1b), indicating increased non-reactive uptake of 13BD by non-acidified seeds which contained no aqueous phase. Figure 16 shows the effect of seed aerosol acidity and RH on the SOC concentrations associated with ER444 and ER 666 experiments.

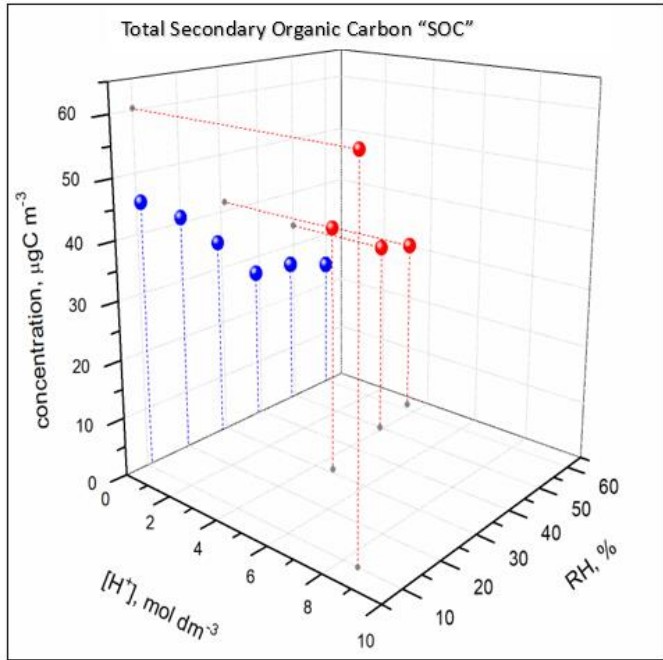

**Figure 16**. Effect of RH and seed aerosol acidity on SOC concentrations under acidified (ER444: red) and non-acidified (ER666: blue) conditions.

The concentration of the majority of reaction products (1) was higher under acidified seed aerosol conditions, where the aerosol particles contained aqueous phase, than under non-acidified seed conditions, where the aerosol particles did not contain any aqueous phase, across the RH range used in this study, (2) increased with the acidity of the aerosol aqueous phase in the experiments under acidified seed conditions, with aqueous phase always present; and (3) decreased with increasing RH either under acidified or non-acidified seed conditions (Figures 12, 13, 14, 17, and S13 – S43). However, there were a few significant deviations. (1) At non-acidified conditions, the concentrations of *meso*-threonic, D-threonic, and dihydroxymaleic acids were higher at all RH values (Figures 17, S23, S25, S36); the concentrations of glyceric acid OS (*m/z* 185 [M-H]$^-$), malic acid OS

($m/z$ 213 [M-H]⁻), and erythrose were higher at RH = (40 – 61)% (Figures 17, S14, S18, S26); and the concentration of glyceric acid-1,3,4-trihydroxy-1-butene ester was higher at RH = (10 – 30)% (Figures 17, S32). (2) In non-acidified experiments, the concentration of 1,3,4-trihydroxy-1-butene was minimal at RH = 50% (Figures 17, S34), and that of malic acid OS ($m/z$ 213 [M-H]⁻) was maximal at RH = 40% (Figure S18). In acidified experiments, the concentrations of glyceric acid-1,3,4-trihydroxy-1-butene ester and 1,3,4-trihydroxy-1-butene were maximal at RH = 30% (Figures. S32, S34) and that of oxalic acid-erythrulose ester was minimal at RH = 30% (Figure S31). The yield trends observed in acidified seed experiments (ER444), in which the aqueous particle phase was always present, indicate that acidity enhances the formation of most products of 13BD oxidation in the aqueous phase. The concentration trends observed in non-acidified seed experiments (ER666), in which the particles contained no aqueous phase, may indicate that the phase state and/or gaseous $H_2O$ limited access of reactants to particle surfaces and hindered the formation of most products. A few exceptions to the major yield trend may indicate more complex formation and partitioning mechanisms involved, which cannot be explained based on current work. In addition, the "acidified" and "non-acidified" terms are associated with seed aerosol solutions, and fundamental differences in chemical mechanism may be responsible for the reaction products formed in this study due the phase states in which the particles themselves exist, liquid (ER444) or solid (ER666). The observed differences between the acidified (ER444) and non-acidified (ER666) seed aerosol experiments may be due to the acidity of the seed aerosol, liquid water content, and/or the phase state of the aerosol particles. Furthermore, more work is needed to gain insight into mechanisms of reactant formation and partitioning to deliquescent particles and to particles lacking the aqueous phase.

The preparation of non-acidified and acidified seed aerosol by nebulizing aqueous solutions of $NH_4SO_4$ and ($NH_4SO_4$ + $H_2SO_4$), respectively, has been used in chamber experiments for many years (Czoschke et al., 2003; Deng et al., 2021; Zhang et al., 2023). As stated above, the LWC and acidity of the aerosol were estimated using E-AIM modeling, which assumes the vapor-particle partitioning equilibrium in the system. The non-acidified seed aerosols were dry based on E-AIM modelling. This may be a major difference in the acidified and non-acidified conditions and probably one of the main drivers of differences in the chemistry between the two systems. Note that E-AIM is an equilibrium model, which has its own limitations. Therefore, it is important to consider that under our experimental conditions, the system should be at chemical equilibrium, which can be difficult to characterize. In our experimental system, we are going from liquid solution in the nebulizer reservoir, to ammonium sulfate particles in near -100% RH air exiting the nebulizer to chamber dry air in less than a minute. Then the particles have a chance to equilibrate in the chamber for a matter of hours. The equilibrium time needed for acidified wet aerosols to adjust their LWC to the chamber RH conditions is probably on the order of minutes to ten of minutes (Saleh et al. 2013), so they are probably close to their E-AIM predicted LWCs in the chamber. However, the time needed for the wet neutral aerosols to completely dry out at room temperature may be on the order of hours. It is probable that under our experimental conditions the wet neutral aerosol do not reach the E-AIM predicted dryness on the time scale of the chamber residence time. This may explain why the two systems are not drastically different chemically as expected.

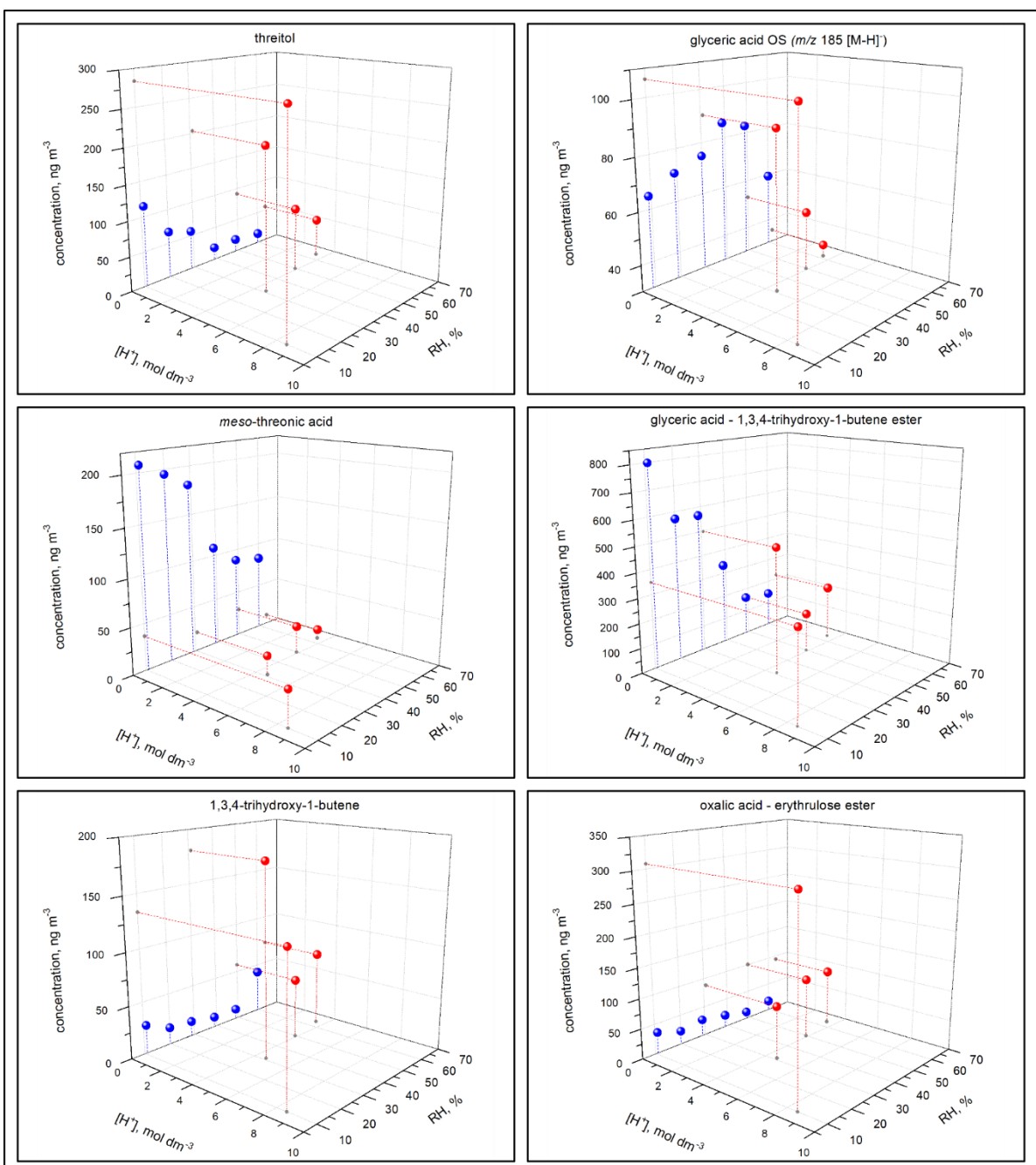

**Figure 17**. Effect of RH and seed aerosol acidity on the concentration of threitol, glyceric acid OS (*m/z* 185 [M-H]⁻), *meso*-threonic acid, glyceric acid – 1,3,4-trihydroxy-1-butene ester, 1,3,4-trihydroxy-1-butene ester, and oxalic acid-erythrulose ester produced in non-acidified (blue) and acidified (red) experiments.

## 3. 4 Field Measurements

We compared the results of chamber experiments conducted under acidified (ER444) and non-acidified (ER666) conditions with ambient PM$_{2.5}$ samples collected at Godów, Zielonka and Diabla Góra monitoring stations in Poland (section 2.2). The purpose of presenting the field results was to show that some compounds observed in smog-chamber experiments occur also in the real ambient aerosol. Four products, which did not contain sulfate groups, were detected both in the chamber SOA and ambient samples: tartronic acid (MW 120), malic acid (MW 134), threonic acid (MW 136) and tartaric acid (MW 150). The same analytical LC-MS method was used for the chamber and field samples. Figure 18 shows the extracted ion chromatograms associated with the ER444 and ER666 chamber experiments, and with the PM$_{2.5}$ samples collected during 2014 and 2016 field studies (section 2.2). Several species (e.g., glyceric acid, malic acid, threitol, erythritol, and tartaric acid) were previously observed in ambient samples collected at several places around the U.S.A. and were also associated with 13BD oxidation (Jaoui et al., 2014). From this work, it now appears that these compounds could be generated in the atmosphere from the oxidation of 13BD in another part of the world (Poland).

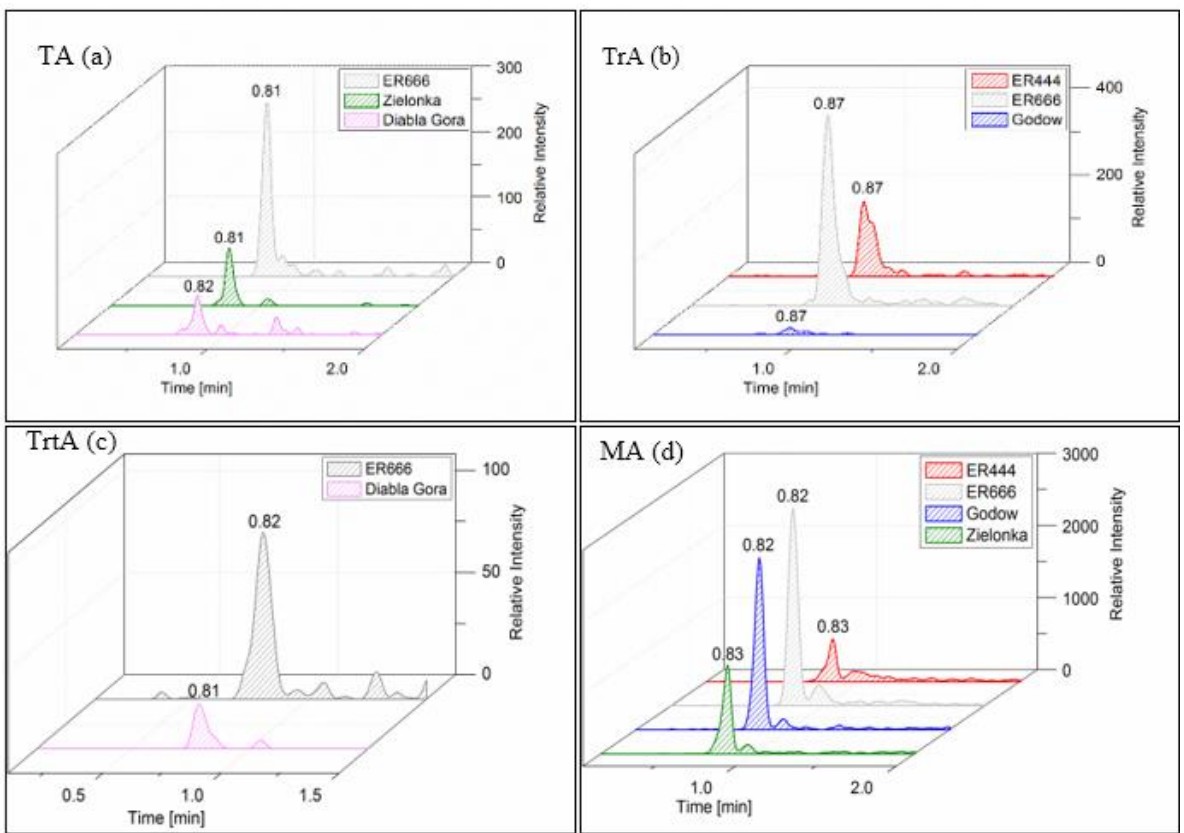

**Figure 18**. Extracted Ion Chromatograms (EICs) recorded for samples from field studies (Zielonka, Godów, and Diabla Góra) and smog chamber experiments (ER444 and ER666): (a) of tartaric acid (TA, MW 150); (b) threonic acid (TrA, MW 136); (c) tartronic acid (TrtA, MW 120), and (d) malic acid (MA, MW 134).

Note that samples from chamber experiments conducted in our laboratory involving the oxidation of biogenic (e.g., isoprene, monoterpenes, sesquiterpenes, 2-methyl-3-buten2-ol), aromatic (e.g., toluene, 1,3,5- trimethylbenzene, benzene), and polycyclic aromatic hydrocarbons (PAHs) (e.g., naphthalene) did not contain the compounds identified in this section except malic acid and tartaric acid using the same experimental and analytical procedures. A second comparison of chamber SOA samples form ER444 and ER666 experiments with ambient PM$_{2.5}$ samples was conducted using the LC-MS analysis focused

on OSs. Extracted ion chromatograms in Figure 10 show that several OSs occurred in chamber SOA and ambient PM$_{2.5}$, including glyceric acid-organosulfate; 2(3$H$)-furanone, dihydro-3,4-dihydroxy-organosulfate (2 isomers); malic acid-organosulfate; and threonic acid-organosulfate.

Interestingly, some organic compounds observed in the gas phase in chamber experiments (Figure 1, Table 3), were reported to be ubiquitous in ambient air. High concentrations of formaldehyde, and to a lesser extent glyoxal, acrolein, PAN and APAN were observed from 13BD oxidation under both acidified and non-acidified seed aerosol across the whole RH range considered in this study. These compounds are ubiquitous in urban ambient samples so 13BD may be their precursor in areas with considerable 13BD emission. The presence of secondary formaldehyde and acrolein (S-HAPs) and the effect of seed aerosol acidity and RH on their formation highlight the importance of this study mainly for health effect studies.

## 4 Summary

While it is impossible to cover and analyze all reaction products associated with 13BD oxidation, this study addresses many of the main products that form under various acidified and non-acidified seed aerosol and relative humidity conditions. We identified more than 50 oxygenated organic compounds in the gas and particle phases obtained from chamber oxidation of 13BD. Some reaction products have been reported in the literature, but several others were identified for the first time: glyceric The yield trends observed in acidified acid-organosulfate, 2(3$H$)-furanone, dihydro-3,4-dihydroxy-organosulfate (2 isomers), threonic acid-organosulfate, 1,2,3,4-butanetetrol nitrosoxy-organosulfate, 1,2,3,4-butanetetrol nitroxy-organosulfate, malic acid-organosulfate, and threonic acid nitroxy-organosulfate. The quantitative analysis showed that glyceric acid, threitol, erythritol, threonic acid, several dimers, unknowns, and organosulfates were the most abundant components of 13BD SOA under acidified and non-acidified seed conditions. Other components contributing to SOA mass were diols, mono- and dicarboxylic acids, organosulfates, dimers, and nitroxy- and nitrosoxy-organosulfates. Several carboxylic acids, organosulfates and nitroxy-organosulfates identified in chamber samples were also detected in ambient aerosol samples collected at various sites in Poland. Such consistency reinforces the relevance of the chamber findings even though some components were found only in chamber experiments.

Comparison between experiments conducted with acidified and non-acidified seed aerosol at various RH revealed that acidity of seed aerosol (ER444), aerosol phase state (ER444 vs. ER666), and RH (ER444 and ER666) influenced the production of SOA as well as the number and molecular structure of products formed in the gas and particle phases. Total SOC decreased with increasing RH, but the levels of individual components varied, though the majority followed similar trends as the SOC. Ozone production and conversion efficiency of 13BD depended on RH, aerosol phase state, and acidity of the seed aerosol similar to SOC. The concentrations of most compounds decreased as RH increased, but few products were most abundant at intermediate RH around 30% (e.g., malic acid, glyceric acid-OS). This is also true for gas phase formaldehyde and acrolein, whose yields increased significantly at higher RH. However, at high humidity, the difference was relatively small. For the range of RH considered, the acidified seed experiments, in which the aerosol particles were deliquescent, enhanced SOC production compared to non-acidified seed experiments, in which the aerosol particles did not contain aqueous phase. SOC production increased with the acidity of the aqueous phase in the acidified seed experiment ER444. Yields of the individual SOA components followed the same pattern as SOC, while a few were more abundant in non-acidified seed experiments (e.g., unknowns) or behaved in an inconsistent manner. Although, the terms "acidified" and "non-acidified" is true for the solutions from which the seeds were atomized, there is far more fundamental differences between the phase states in which species partition to or from (aqueous/solid), which considerably affects their partitioning and formation mechanisms. Therefore, further research is needed to elucidate the mechanisms of their formation and partitioning in the atmosphere, both for dry and deliquescent particles. Overall, the 13BD results are consistent with those reported by Nestorowicz et al. (2018) for isoprene.

The present study employed analytical methods suitable for a wide range of 13BD reaction products in the particle and gas phases at various RH and seed aerosol acidity. It provided a solid identification of wide variety of reaction products including HAPs, dimers and organosulfates, as well as quantitative changes in their formation as a function of RH and seed aerosol acidity. Certainly, this study includes some uncertainties that need to be addressed in future work. Perhaps the most significant improvement would be the use of authentic standards to better quantify the contribution of individual products to SOA at various RH and seed aerosol acidity. This study shows that SOA and SOC consistently increased with decreasing RH and were higher under acidified than non-acidified seed conditions. Similar trends were observed for the majority of reaction products

reported in this study. To assess that observation, the LWC and other thermodynamic properties of the aerosol phase were estimated as a function of RH using the thermodynamic model E-AIM. Organic acids could have more profound influence, especially for the non-acidified seed experiment, so the results of our modeling should be taken as a first approximation and verified as soon as the precise stoichiometry of the acids formation and reliable dissociation constants of those acids are available.

13BD does not contribute more to atmospheric aerosol than isoprene, which is emitted to the atmosphere in a much larger amount and plays a significant role in atmospheric SOA modelling. However, 13BD emissions are less understood than isoprene, and current increases in wildfires and acreages burned due to climate issues can increase 13BD emissions and therefore the production of 13BD SOA, which may have implications for air quality, health, and well-being.

*Data Availability.* The data used in this study can be found at https: //www.data.gov/. All data presented in this paper can be requested by contacting the corresponding author.

*Author contributions.* MJ: study conceptualization; standards and chamber sample analysis; data collection; result interpretation; writing original and revised drafts; communication. ML: conducted chamber experiments; data collection and analysis. TEK: supervision; study conceptualization; result interpretation; editing drafts. KN, KJR, JT, EB, WD, and RS: LC-MS analysis, result interpretation; editing drafts.

*Competing interests.* The authors declare no competing financial interest.

*Disclaimer.* This work has been subjected to the U.S. Environmental Protection Agency's administrative review and approved for publication. The views expressed in this article are those of the authors and do not necessarily represent the views or policies of the U.S. Environmental Protection Agency. Mention of trade names does not constitute endorsement or recommendation of a commercial product by U.S. EPA. The work of Polish researchers, i.e., Rafal Szmigielski and Krzysztof J. Rudzinski, was partially supported by funds from the National Science Centre, Poland, through an OPUS21 grant scheme (No. 2021/41/B/ST10/ 02748).

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
