# Peer review of "Atmospheric oxidation of 1,3-butadiene: influence of seed aerosol acidity and relative humidity on SOA composition and the production of air toxic compounds"

_EGUsphere, 2024_

## Author Comment (AC1)

RC1 review

This is the review for manuscript entitled "Atmospheric oxidation of 1,3-butadiene: influence of acidity and relative humidity on SOA composition and air toxic compounds" by Jaoui et al.

This study investigates the effect of RH on the chemical composition of both gas and particle phases formed from the photooxidation of 1,3-butadiene (13BD) under acidic and non-acidic conditions. The authors identified a variety of products formed through the reaction. They find SOA mass and the most SOA products under acidic and lower RH conditions. With increased RH, secondary organic carbon decreased under acid and non-acid conditions. The authors state that the photochemical reactivity of 13BD in our systems decreased with increasing RH and was faster under non-acidic than acidic conditions. Overall, the results from this study present potential to improve the current understanding of atmospherically relevant aerosol particles. The manuscript is clearly written, and I enjoyed reading it. I mostly suggest minor revisions and clarifications, though a major revision might be needed address an alternative interpretation of the influence of acidity and RH.

Authors' reply:

>We thank the Reviewer for the effort to evaluate our manuscript and for the positive opinion on it.

**General Comments:**

The manuscript discusses the impacts of RH and acidity affects oxidation reactions and products. The changes in RH will modify the acidity of aerosols. It remains unclear how the authors determine the relative influence of those factors. Authors need to clarify their approach in differentiating and quantifying the impact of RH and acidity on SOA formation.

Authors' reply:

>The Reviewer is correct. RH and acidity of aerosols are related for a given aerosol composition. Note though, our experiments were conducted under two conditions: (1) in the absence of acidic seed aerosol (ER666, Table 1) and only the RH was changed from 11 to 60%; (2) under acidic seed aerosol (ER444, Table 1), and then only the RH was changed from 10 to 62%. Then the relative influence of RH can be assessed from both experiments ER444 (under acidic conditions) and ER666 (under non-acidic conditions) as only RH was changed. Although, small acidity changes may occur with changes in relative humidity as indicated by the reviewer, which is happening under acidic and non-acidic conditions (ER444 and ER666), but it is part of the effect of RH also. However, it is difficult to characterize the changes in SOA reaction products, which may be due to dilution, and/or chemical reactions occurring either in the gas or particle phase or both. The effect of RH was discussed in section 3.2.1. The effect of acidity at the same RH was reported in section 3.2.2.

>In addition, to assess that relation in all our experiments, we used the widely used Extended-AIM Thermodynamic Model (E-AIM) as explained in the manuscript, section 2.5 p. 8 lines 15-24. The input data for the model are RH, temperature, and quantitative aerosol composition (Section S2 in the Supplementary).

>The aqueous aerosol phase occurred only in all acidic-seed experiments, where "*The acidity of the aqueous phase decreased, and LWC increased with increasing RH. The pH (aH$^+$) of the aqueous phase increased from -1.79 at 10% RH to ~0.40 at 62% RH*" as we have written in

the original manuscript (Section 3.2.3, p. 29, lines 27-28). Thus, we feel that the comment has already been addressed in the original manuscript.

The study explores a range of RH from 11-60%. At 11% RH, ammonium sulfate particles are likely in the solid phase, which may involve different chemical mechanisms compare to those in the aqueous phase. Authors should consider how the phase states affect aerosol chemistry and the products observed.

Authors' reply:

Again, this is an important point raised by the reviewer. E-AIM modeling showed that the aqueous aerosol phase was absent across all RH conditions in non-acidic seed experiments (Section 3.2.3, p.29, lines 24-25). On the other hand, aqueous aerosol phase occurred in all acidic seed experiments (Section 3.2.3, p. 29, Table 6). Thus, the results of the non-acidic seed experiment refer to non-aqueous chemistry. In contrast, the acidic-seed experiments include the influence of the aqueous-phase chemistry and the acidity of the aqueous aerosol phase. We addressed that difference in the original manuscript (p. 30 lines 16-20): "The yield trends observed in acidic experiments, in which the aqueous particle phase was always present, may indicate that acidity enhances the formation of most products of 13BD oxidation in the aqueous phase. The concentration trends observed in non-acidic experiments, in which the particles contained no aqueous phase, may indicate that gaseous $H_2O$ limited access of reactants to particle surfaces and hindered the formation of most products. A few exceptions to the major yield trend may indicate more complex formation mechanisms involved, which cannot be explained based on current work." Thus, we believe that this comment has been addressed in the original manuscript.

As stated above, based on the E-AIM modelling the non-acidic aerosols were dry. This may be a major difference in the acidic and non-acidic conditions and probably one of the main drivers of differences in the chemistry between the two systems. It is important to note that E-AIM is an equilibrium model, which has its own limitations, and it is out of the scope of this paper. Therefore it is important to note that under our experimental conditions, the system should be at chemical equilibrium, which is difficult to characterize. In our experimental system, we are going from liquid solution in the nebulizer reservoir, to ammonium sulfate particles in near -100% RH air exiting the nebulizer to chamber dry air in less than a minute. Then the particles have a chance to equilibrate in the chamber for a matter of hours. The equilibrium time needed for acidic wet aerosols to adjust their LWC to the chamber RH conditions is probably on the order of minutes, so they are probably close to their E-AIM predicted LWCs in the chamber. However, the time needed for the wet neutral aerosols to completely dry out at room temperature may be on the order of hours. It is probable that under our experimental conditions the wet neutral aerosol do not reach the E-AIM predicted dryness on the time scale of the chamber residence time. This may explain why the two systems are not drastically different chemically as expected.

To reflect the reviewer concern, we added the following paragraph to the manuscript at the end of section 3.2.3, where E-AIM was discussed.

"As stated above, the non-acidic aerosols were dry based on E-AIM modelling. This may be a major difference in the acidic and non-acidic conditions and probably one of the main drivers of differences in the chemistry between the two systems. Note that E-AIM is an equilibrium model, which has its own limitations. Therefore, it is important to consider that under our experimental conditions, the system should be at chemical equilibrium, which is difficult to

characterize. In our experimental system, we are going from liquid solution in the nebulizer reservoir, to ammonium sulfate particles in near -100% RH air exiting the nebulizer to chamber dry air in less than a minute. Then the particles have a chance to equilibrate in the chamber for a matter of hours. The equilibrium time needed for acidic wet aerosols to adjust their LWC to the chamber RH conditions is probably on the order of minutes, so they are probably close to their E-AIM predicted LWCs in the chamber. However, the time needed for the wet neutral aerosols to completely dry out at room temperature may be on the order of hours. It is probable that under our experimental conditions the wet neutral aerosol do not reach the E-AIM predicted dryness on the time scale of the chamber residence time. This may explain why the two systems are not drastically different chemically as expected."

This study did not include the toxicity of compounds formed from SOA; the author may modify the title of the paper to be more relevant.

Authors' reply:

> The revised title is "Atmospheric oxidation of 1,3-butadiene: influence of acidity and relative humidity on SOA composition and hazardous air pollutants."

**Specific Comments :**

**Introduction**

1.      The literature citations need to be updated. Several recent studies indicating how acidity influences SOA formation should be included:

- Decreases in Epoxide-Driven Secondary Organic Aerosol Production under Highly Acidic Conditions: The Importance of Acid–Base Equilibria Madeline E. Cooke, N. Cazimir Armstrong, Alison M. Fankhauser, Yuzhi Chen, Ziying Lei, Yue Zhang, Isabel R. Ledsky, Barbara J. Turpin, Zhenfa Zhang, Avram Gold, V. Faye McNeill, Jason D. Surratt, and Andrew P. Ault Environmental Science & Technology 2024 58 (24), 10675-10684 DOI: 10.1021/acs.est.3c10851

- Initial pH Governs Secondary Organic Aerosol Phase State and Morphology after Uptake of Isoprene Epoxydiols (IEPOX) Ziying Lei, Yuzhi Chen, Yue Zhang, Madeline E. Cooke, Isabel R. Ledsky, N. Cazimir Armstrong, Nicole E. Olson, Zhenfa Zhang, Avram Gold, Jason D. Surratt, and Andrew P. Ault Environmental Science & Technology 2022 56 (15), 10596-10607 DOI: 10.1021/acs.est.2c01579Method

Authors' reply:

> The references are now incorporated on page 3, line 23 The references were added to the reference section.

**Method**

1.      The manuscript should address the wall loss of particles within the smog chamber. what is the residence time of particles in the smog chamber?

Authors' reply:

For a wide range of particles, the wall loss rate is 0.063 h$^{-1}$.

We added the following sentences on page 5, line 28 (original manuscript).

"For a wide range of particles, the wall loss rate is 0.063 h$^{-1}$. However, with the chamber running in a flow mode the wall loss rate is subsumed in the observed decrease from the input reactants and the steady-state concentrations."

The residence time of 4 hour was reported in the manuscript in section 2.1.

2.      The manuscript does not mention the concentration of ammonium sulfate and sulfuric acid solutions under acidic and non-acidic conditions.

Authors' reply:
We have included that information in the Supplementary (Section S2. p. 19).

3.      It is unclear whether each experiment started at the lowest RH, approximately 11%. If so, the ammonium sulfate seed particles would be below the efflorescence point and remain in the solid phase across the RH range, influencing SOA formation differently compared to reactions in the liquid phase. Can the authors confirm and discuss this?

Authors' reply:
We showed by E-AIM modeling that in all non-acidic seed experiments (RH = 11-60%) there was no aqueous phase while in all acidic-seed experiments (RH = 10 – 62%) the aqueous phase occurred. As noted above, the results of non-acidic seed experiment refer to non-aqueous chemistry while the acidic-seed experiments include the influence of the aqueous-phase chemistry and the acidity of the aqueous aerosol phase (original manuscript, p. 30 lines 16-20). Please refer to our response to your general comments.

4.      Changes in RH also affect aerosol acidity. How do the authors assess the impact of pH changes along with RH adjustments? Furthermore, aerosol size significantly affects aerosol pH. Were the inorganic seed particles size-selected before being introduced into the smog chamber? An explanation on how size effects on aerosol acidity were ruled out would be beneficial.

Authors' reply:
We addressed the relation of RH and aerosol acidity in the previous parts of our response. The influence of aerosol size on aerosol acidity generally refers to atmospheric aerosols and is controlled by chemical composition, variation of inorganic aerosol composition with size, hygroscopicity, and timescale differences of gas-particle equilibration (mass transfer). The seed aerosol in smog-chamber experiments is prepared and has a defined chemical composition so its acidity does not depend on particle size. Formation and partitioning of reaction products, especially organic acids, to seed-aerosol particles could affect the aerosol acidity, but we neglected that influence because we did not know the exact stoichiometry of dissociable hydrogen ions production during their formation as well as reliable dissociation constants. We reported that in the original manuscript (p.8, Section 2.5, lines 21-23).

While we appreciate the reviewer's comments, the effect of particle size on the results was not studied and this consideration is out of the scope of this study.

5.      How many experiments were conducted under the various conditions, and how repeatable are the results?

Authors' reply:

We have conducted more than 10 experiments of this particular experimental design, and the experiments are reproductible within the experiment's uncertainties. Though it is impossible to repeat experiments exactly, good reproducibility has been achieved vis-a-vis the aerosol yield and the compounds observed. Repeatability of RH conditions was quite good. Repeatability of seed aerosol conditions was more challenging, which was why these experiments involved changing the RH while the seed was kept constant rather than holding the RH constant and changing the seed.

6.      The RH at the ambient sampling sites is consistently higher than that in the chamber conditions. Why were higher RH experiments not performed in this study?

Authors' reply:

For the steady-state operating mode used for these experiments, the highest stable RH level we could achieve was ~65% RH.

The primary purpose of presenting the composition of ambient samples was to show that some compounds observed in the chamber also occurred in the ambient aerosol. We did not attempt to compare the real and chamber processes quantitatively.

**Discussion**

1.      Can the authors explain why the photochemical conversion of 13BD decreases with an increase in RH? The conversion efficiency of 13BD is higher under non-acidic conditions compared to acidic conditions. As the RH increases and the acidity decreases, this should logically result in an increase in 13BD conversion efficiency. Could the authors discuss this observation?

Authors' reply:

We did provide explanation for "The photochemical reaction of 13BD in the presence of NOx decreases with an increase in RH" on page 8, lines 40-41. However, to reflect the reviewer concern, we added the following sentences to page 8, line 47 and read:

"It has been reported that RH changes promote changes in OH radical concentrations (Hu et al., 2011). These changes may be one of the reasons for the RH affecting the photochemical conversion of 13BD in our study."

We added the following reference to the reference section:

Hu, G., Xu, Y., and Jia, L.: Effects of relative humidity on the characterization of a photochemical smog chamber, J. Enviro. Sci. 23, 2013–2018, 2011.

We added also the following sentence on pages 29-30 (lines 40 and 1, respectively):

"On the contrary, the 13BD conversion efficiency was higher under non-acidic conditions than acidic (Figure 1b), indicating increased non-reactive uptake of 13BD by non-acidic seeds which contained no aqueous phase."

The reviewer states that as "RH increases and the acidity decreases, this should logically result in an increase in 13BD conversion efficiency". We don't have any experimental evidence of this statement (RH increases can lead to acidity decreases) as we did not measure experimentally and don't have the capability in our lab to measure the pH of the aerosol. Although our data suggest acidic compounds decreased with increases RH (Figure 12: top panel), we do not have any evidence that increases of RH lead to decreases in aerosol acidity (dilution effect). There is probably others chemistry be involved (e.g. heterogenous chemistry), and it is out of the scope of this study.

2.      It is interesting to observe the concentration of the majority of products decreased with increasing RH, can author explain why?

Authors' reply:

We have suggested the possible explanation in the original manuscript (Section 3.2.3, p. 30, lines 16-20). In the acidic-seed experiments, the aerosol acidity decreased with increasing RH, so we suggested that acidity enhances the formation of most products in the aqueous-phase. In the non-acidic-seed experiments, we suggested that formation of gaseous $H_2O$ limited the access of reactants to aerosol surfaces and hindered the formation of most products.

3.      Authors should consider add E-AIM result for non-acidic particles at different RH conditions for comparison and further understand the effect of acidity on SOA formation.

Authors' reply:

We have done E-AIM modeling for non-acidic seed aerosol and described the results in the original manuscript (Section 3.2.3, p. 29, lines 23-25). In all non-acidic seed experiments, there aerosol aqueous phase was absent.

4.      Page 24, Line 22, Need to add unit after 45.1.

Authors' reply:

We added the unit as suggested by the reviewer.

5.    Authors found that the higher concentration of the majority of reaction products under acidic conditions; however, it is not clear that this result is due to the effect of acidity, RH, or the phase state of seed particles.

Authors' reply:

We discussed this in our response to the first comment in the general comments of this reviewer. As we discussed in our original paper (section 3.2.2), there is clear evidence that acidity affects reaction products formation. We believe that this comment is discussed in our paper (section 3.2.2). Please note that the concentrations are in mass units per cubic meter of air. We respectfully acknowledge that the present work cannot explain the chemistry involved in our experiments better and that further work is needed (original manuscript, p. 30, lines 19-20; p. 33, lines 31-33).

---

## Author Comment (AC2)

RC2 review

In "Atmospheric oxidation of 1,3-butadiene: influence of acidity and relative humidity on SOA composition and air toxic compounds," Jaoui et al. analyzed the composition of SOA prepared from 1,3-butadiene in the presence of $NO_x$ under varying acidity and humidity levels, including both the gas and particle phases. They found increased production of SOC at elevated RH and increased acidity, although some individual compounds varied from these trends. They also identified and reported several compounds from the oxidation of 13BD for the first time. The analysis presented was in-depth and informative, and worthy of publication after the following points are addressed.

Authors' reply:

> We thank the Reviewer for the effort to evaluate our manuscript and for the positive opinion on it.

Major Feedback

1. The stated goals of the paper are not the major points of discussion throughout the work. For example, the authors state the purpose of the paper is to analyze the impacts of LWC on SOA formation, but LWC is not discussed in the paper other than Page 29 line 7 where it is related to acidity. I think LWC warrants more discussion, particularly for the non-acidified seed particle experiments where some of the seed particles are likely effloresced. I would expect a large difference in chemical behavior between solid and aqueous particles.

Authors' reply:

> LWC estimated using the E-AIM model was zero in all non-acidic experiments as written in the original manuscript (p. 29 line 24), so it could not influence the results. In the acidic experiments, LWC increased proportionally with RH (Table 6 in the original manuscript and the plot below), so the influence of LWC is equivalent to that of RH (in the acidic experiments only).

[Figure]

> We provided additional discussion in our response to reviewer #1 to his general comments. (please refer to that comment for additional info).

2. Many of the identified products appear to be OH oxidation products judging by retention of the 4-carbon backbone. Do the authors have an estimate of the steady state OH concentration? Does it change with any of the parameters they are varying (perhaps with LWC)? Given the importance of OH oxidation in the atmosphere, and that it is the major atmospheric sink for isoprene, the impact of the variables on OH concentrations should be considered.

Authors reply:

The reviewer makes a good point. This is correct that OH radicals are the main oxidant in our system both under acidic and non-acidic conditions. Unfortunately, we don't have the capability in our lab to measure OH radicals in the gas phase.

We provided additional discussion concerning this comment in our response to reviewer #1 (section Discussion; comment #1). In that response we added the following sentences to the manuscript (please refer to that comment for additional info).

"It has been reported that RH changes promote changes in OH radical concentrations (Hu et al., 2011). These changes may be one of the reasons for the RH affecting the photochemical conversion of 13BD in our study."

3. If possible, the section on field measurements should be strengthened. The results would be more meaningful if 13BD had also been measured at the sites, but from my understanding there were only particle measurements taken in the field. It is stated in the intro that 13BD is mostly from anthropogenic sources, and then it is stated that the site in Godow has more anthropogenic influence than the other sites. Were the 13BD products higher there than in the other sites, or is there other evidence that the concentrations of proposed 13BD oxidation products are higher in areas where higher levels of 13BD are expected to be found?

Authors' reply:

In our field campaigns, we collected the ambient aerosol samples only and relied on the routine measurements from the monitoring stations at which we sampled. Unfortunately, 13BD is not routinely measured at Polish monitoring stations. However, the purpose of presenting the field results was to show that compounds determined in smog-chamber experiments occur also in the real ambient aerosol. The quantitative analysis of those compounds in the atmosphere at specific locations was far beyond the scope of our work.

To reflect the reviewer concern, we added the following sentence to the revised manuscript on page 32, line 2 and reads:

"The purpose of presenting the field results was to show that some compounds observed in smog-chamber experiments occur also in the real ambient aerosol."

Minor Feedback

Page 3 Line 25-30 - This statement could be clarified or perhaps made more specific. What is meant by "bulk properties" versus "composition?" Several of the listed citations for only "bulk properties" show analysis of composition. I think perhaps the authors are trying to differentiate between studies that looked at only particle composition versus studies that also included the gas phase composition. There have been several further studies on the effects of RH on particle composition since 2021.

For example:

https://doi.org/10.1039/D3EA00033H Klodt

https://doi.org/10.1039/D3EA00149K luo

https://doi.org/10.1039/D3EA00128H Thomsen

There are likely others that I did not find in my brief search, and the earlier studies in this collection of work may be better represented by a review paper if one is available. Also, the year for the Hinks reference should be 2018, rather than 1918.

Authors' reply:

We refer to SOA "bulk properties" those properties that are associated with macro-parameters of the aerosol such as yield, SOC, SOA of the aerosol.

We added references to the revised manuscript as suggested by the reviewer. See also specific comment #1 from reviewer # 1, specific comments)

1. Klodt, A. L., Aiona, P. K., MacMillan, A. C., Lee, H. J., Zhang, X., Helgestad, T., Novak, G. A., Lin, P., Laskin, J., Laskin, A., Bertram, T. H., Cappa, C. D., and Nizkorodov, S. A.: Environ. Sci.: Atmos., Effect of relative humidity, NOx, and ammonia on the physical properties of naphthalene secondary organic aerosols, 3, 991, doi.org/10.1039/D3EA00033H, 2023.
2. Luo, H., Guo, Y., Shen, H., Huang, D. D., Zhangabcef, Y., and Zhao, D.: Effect of relative humidity on the molecular composition of secondary organic aerosols from a-pinene ozonolysis, Environ. Sci.: Atmos., 4, 519, doi.org/10.1039/D3EA00149K, 2024.
3. Thomsen, D., Iversen, E. M., Skønager, J. T., Luo, Y., Li, L., Roldin, P., Priestley, M., Pedersen, H. B., Hallquist, M., Ehn, M., Bilde, M., and Glasius, M.: The effect of temperature and relative humidity on secondary organic aerosol formation from ozonolysis of D$_3$-carene, Environ. Sci.: Atmos., 4, 88, doi.org/10.1039/D3EA00128H, 2024.

Page 5 line 2 - Does all of the radiation fall between 300 and 400 nm? It could be useful to provide a graphical comparison between the lamp spectrum and the solar spectrum in the SI, unless it is provided in one of the references.

Authors' reply:

As stated in our manuscript (page 5, lines 1, 2), the UV fluorescent bulbs used in the chamber produce radiation in the actinic region of the spectrum at 300-400 nm photolytically comparable to that of solar radiation (Black et al., 1998). The readers should consult Black et al., 1998 paper.

Page 5 line 6 - What was the approximate pH of the non-acidified seed aerosol? I would expect it to be between 4 and 5, so "non-acidified" would be a more accurate description of the seeds than "non-acidic" The EAIM results for the non-acidified conditions should also be included in the paper. As the dissolved organic acids will have a significant effect on the pH for the unacidified conditions, the

authors could also consider adding some concentration of a representative organic acid to the calculations to get a general idea of the pH of the particles during these experiments.

Authors' reply:

> The results of E-AIM modeling of non-acidic seed experiments have been included in the original manuscript (p. 29, lines 24-25; input data in Supplementary, p. 19) – there was no aqueous phase in all non-acidic experiments. The influence of dissolved organic acids produced from 13BD was not considered as explained in the original manuscript (p. 29, lines 28-34). It could be marginal in acidic-seed experiments due to dominating inorganic acidity. It could be more profound in non-acidic experiments, but we were not able to evaluate it quantitatively because of lacking stoichiometric and dissociation data.

> We believe the reviewer refers mainly to the terminology. Acidified aerosol means a laboratory prepared acidic seed aerosol introduced into the chamber initially and the actual pH of the wet seed without added acidification from acids present in the aerosol phase. For example, if one adds the wet aerosol from the nebulizer to the chamber at an RH greater than the efflorescence RH (ERH) of ammonium sulfate (ca. 30%), the seed aerosol will stay at a liquid state. This is the condition the reviewer is referring to. If non-acidified aerosol in introduced to the chamber with an RH < 30%, then the aerosol will not retain its wetness.

Table 1 - What is the significance of the experiment set names? I found them difficult to remember on initial read through. Since they are currently separated by acidity level, could they be referred to using an abbreviation that correlates to their acidity level?

Authors' reply:

> We thank the reviewer of this comment. We chose to keep these names for quality controls purpose. These names are associated with the original experiment's names.

Page 10 Line 6 - Are the authors referring to wall loss here? Or is there another type of loss they anticipate?

Authors' reply:

> We thank the reviewer for this comment. Yes, we refer to wall loss here. We changed the following sentence (page 10, line 6) from:

> "Given the experiments were conducted in flow mode, we anticipate the loss in particle and gas phase products including 13BD to be negligeable as the measurements were conducted at steady state conditions."

> To

> "The wall loss of particles is ~6% per hour. We expect gas phase products and 13BD wall to be negligeable as the experiments were conducted at steady state conditions, and in flow mode."

Table 4 – It would be informative if the authors could provide a ratio of intensity in acidified to unacidified conditions (or vice versa). They would then have the option to comment on the relative production of the compound under each condition.

Authors' reply:

> We are not certain what the reviewer is referring to here as "intensity." Is this a reference to GC signal intensities (or estimated concentrations)? If so, this ratio would provide a good indication of the differences in abundance of the identified compounds under the different conditions tested. However, this would not be straightforward to do, because we have multiple RH conditions as an additional dimension in these measurements, which would make a simple ratio unworkable, or at least much more complex than a single ratio per compound.

Figure 6 - The authors show the less acidic carboxylic acid being deprotonated for oxaloacetic acid. This structure would make the decarboxylation they show less favorable. The fragmentation pathway for this intermediate has been presented before in acidified and unacidified ammonium sulfate solutions under conditions relevant to this work: https://doi.org/10.1021/acsearthspacechem.1c00025

Authors' reply:

> If we understand the reviewer's comment correctly, that one of the two structures we proposed for the $m/z$ 87 ion, i.e., the product ion from the decarboxylation of deprotonated oxaloacetic acid precursor ion ($m/z$ 131) is less favorable. In Figure 6 we proposed the two possible structures for $m/z$ 87 ion:
>
> First:

> which is consistent with the Klods' paper – stating that the decarboxylation proceeds through the proposed scheme, which include a six-membered ring transition state (see Scheme 2 in the

> Klod's paper):
>
> However, the second structure we proposed is also possible:

> In contract to solution chemistry, our results presented in Scheme 6 refer to fragmentation reactions in the negative electrospray mass spectrometry. These gas-phase reactions cover not only simple C-C bond breaking, which are typical of solution chemistry but also rearrangements, where two bonds may decompose simultaneously giving rise to an isobaric product. The reviewer states that the second proposed structure is "less favorable". However,

the formation of latter could be easily explained through the ion-neutral complex-mediated reactions, which are recognized mechanisms in mass spectrometry (Longevialle, Mass Spectrom Rev., 1992, 11, 157-192). As shown below, the m/z 87 ion may decompose through the negative charge-initiated fission of the adjected C-C bond resulting in the formation of a short living ion neutral complex 1. While travelling to the detector, such a complex may decompose leading to m/z 43 product ions (not observed in our mass spectra in Figure 6) but also the 1,2-H shift reaction may occur between the neutral molecule ($CO_2$) and a charged fragment of the complex, which leads to a novel ion-neutral complex 2. Once it is formed, the carboxylation happens leading to isobaric structure of the *m/z* 87 ion that we postulated in Figure 6:

*m/z* 87        ion-neutral complex 1        ion-neutral complex 2

*m/z* 87

References

Alexandra L. Klodt; Kimberly Zhang; Michael W. Olsen; Jorge L. Fernandez; Filipp Furche; and Sergey A. Nizkorodov. Effect of Ammonium Salts on the Decarboxylation of Oxaloacetic Acid in Atmospheric Particles. *ACS Earth Space Chem.* 2021, 5, 4, 931–940.

Longevialle, Pierre. Ion–neutral complexes in the unimolecular reactivity of organic cations in the gas phase. Mass Spectrom Rev., 1992, 11, 157-192.

Page 32 lines 18 through 20 - When the authors say the compounds they identified in this work were not found in SOA from other precursors, do they mean just the ones identified in Tables 4 and 5, or just the 4 compounds listed earlier in the section? Because several of the compounds mentioned in the paper, for example formaldehyde, should be common oxidation and photolysis products.

Authors' reply:

We thank the reviewer for this comment. To reflect his concern, we changed the following sentence (page 32) from

"Note that samples from chamber experiments conducted in our laboratory involving the oxidation of biogenic (e.g., isoprene, monoterpenes, sesquiterpenes, 2-methyl-3-buten2-ol), aromatic (e.g., toluene, 1,3,5- trimethylbenzene, benzene), and polycyclic aromatic hydrocarbons (PAHs) (e.g., naphthalene) did not contain the compounds identified in work using the same experimental and analytical procedures."

To

" Note that samples from chamber experiments conducted in our laboratory involving the oxidation of biogenic (e.g., isoprene, monoterpenes, sesquiterpenes, 2-methyl-3-buten2-ol), aromatic (e.g., toluene, 1,3,5- trimethylbenzene, benzene), and polycyclic aromatic hydrocarbons (PAHs) (e.g., naphthalene) did not contain the compounds identified in this section except malic acid and tartaric acid using the same experimental and analytical procedures."

---

## Author Response (AR3)

**UNITED STATES ENVIRONMENTAL PROTECTION AGENCY**

**Center for Environmental Measurements & Modeling**

**Research Triangle Park, NC  27711**

November 01, 2022

**\*Comments by the Editor are in blueprint, answers by the authors are in normal print**

1. As both reviewers noted, this manuscript presents and thorough and comprehensive analysis, which is highly informative about the chemical details of the aerosol system investigated. Both reviewers suggested publication after minor revisions, and I agree that the article will provide valuable insight for the community on chemical processes in organic aerosol formation.

Authors' reply:

> We thank the Editor for the effort to evaluate our manuscript and our response to reviewers' comments and for his positive opinion on the present manuscript.

2. However, both reviewers of this manuscript rightly pointed out that the interpretations of data trends are sometimes confusing, especially with regard to comparison between hydrated and solid seed particles. Throughout, these are referred to as "acidic" (or acidified) and "non-acidic", which is true of the solutions from which the seeds were atomized, but the far more fundamental difference between experiment sets 444 and 666 is that the particles themselves exist in different phases, which drastically changes the ability of gas-phase species to partition to the particle phase. This difference in phase is not mentioned in the abstract or conclusions; instead, there (and throughout, e.g. in sections 3.2.2 and 3.2.3), changes in the organic aerosol composition between the two experiments are attributed to particle acidity. But it cannot be stated conclusively that observed differences are "dependent on" particle acidity, as that wasn't the only difference between the 444 and 666 experiments -- phase state was as well, and these weren't separated. (Indeed it's questionable whether the non-acidified experiments can even be referred to as "less acidic" if they don't have an aqueous phase and therefore don't have a definable "acidity", so there isn't really a comparison to a less acidic solution being made.

Authors' reply:

Throughout the paper we pointed out that the aerosol aqueous phase occurred in all acidified seed experiments but was absent in all non-acidified seed experiments. However, we agree with the Editor assessment that additional details are needed, and the abstract and summary did not contain the adequate information on the phase state of aerosol particles in the two types of experiments. We also clearly clarified that between ER444 and ER666 experiments, it was the seed aerosol acidity and not the aerosol acidity which affected the formation of SOA in this system. To reflect the Editor concern, we changed the abstract from:

[revised manuscript text omitted]

We replaced the word "acidic" by "acidified" throughout the manuscript and SI including figures and Tables to reflect the editor concerns. We also make sure that the term "acidified" and "non-acidified" is associated with the seed aerosol solution.

The title was changed from:

"Atmospheric oxidation of 1,3-butadiene: influence acidity and relative humidity on SOA composition and hazardous air pollutants"

To

"Atmospheric oxidation of 1,3-butadiene: influence of seed aerosol acidity and relative humidity on SOA composition and the production of air toxic compounds"

3. If experiments were performed that isolated the effects of acidity without an accompanying change in phase state, such conclusions could be drawn, but without that, the effects of acidity are not separable from those of phase state.

Authors reply:

We completely agree with the editor on this comment. However, our experiment ER444 presents a series of stages at changing RH (therefore changing acidity due to dilution) but without a change in phase state (all stages in ER444 have deliquescent particles). ER444 shows that in most cases acidity promoted formation of the products. To reflect the editor concern and make sure that this statement is exclusively associated with experiment ER444, we changed the following paragraph (page 31, lines 17: re-revised manuscript) from:

"The yield trends observed in acidic experiments, in which the aqueous particle phase was always present, may indicate that acidity enhances the formation of most products of 13BD oxidation in the aqueous phase. The concentration trends observed in non-acidic experiments, in which the particles contained no aqueous phase, may indicate that gaseous $H_2O$ limited access of reactants to particle surfaces and hindered the formation of most products. A few exceptions to the major yield trend may indicate more complex formation mechanisms involved, which cannot be explained based on current work."

To

> "The yield trends observed in acidified seed experiments (ER444), in which the aqueous particle phase was always present, indicate that acidity enhances the formation of most products of 13BD oxidation in the aqueous phase. The concentration trends observed in non-acidified seed experiments (ER666), in which the particles contained no aqueous phase, may indicate that the phase state and/or gaseous $H_2O$ limited access of reactants to particle surfaces and hindered the formation of most products. A few exceptions to the major yield trend may indicate more complex formation and partitioning mechanisms involved, which cannot be explained based on current work. In addition, the "acidified" and "non-acidified" terms are associated with seed aerosol solutions, and fundamental differences in chemical mechanism may be responsible for the reaction products formed in this study due the phase states in which the particles themselves exist, liquid (ER444) or solid (ER666)."

To reflect the Editor's concern, we also changed the following sentences (page 31 line 5: re-revised manuscript) from:

> "The concentration of the majority of reaction products (1) was higher under acidic conditions, than under non-acidic conditions, across the RH used in this study, (2) decreased with increasing RH either under acidic or non-acidic conditions (Figures 12, 13, 14, 17, and S13 – S43). "

To

> "The concentration of the majority of reaction products (1) was higher under acidified seed aerosol conditions, where the aerosol particles contained aqueous phase, than under non-acidified seed conditions, where the aerosol particles did not contain any aqueous phase, across the RH range used in this study, (2) increased with the acidity of the aerosol aqueous phase in the experiments under acidified seed conditions, with aqueous phase always present; and (3) decreased with increasing RH either under acidified or non-acidified seed conditions (Figures 12, 13, 14, 17, and S13 – S43)."

4. The authors pointed out in their response to reviews that particle phase state was discussed in the original manuscript. However, the fact that both reviewers remained troubled by the interpretations, and that the manuscript's data interpretation point conclusively to the effects of acidity despite not separating the confounding effects of phase state, means that the original discussion of phase state in the paper was insufficient. I'd therefore like to ask that the authors further revise the manuscript to clarify, throughout, that the observed differences *may* be due to acidity, but alternatively *may* be due to phase state or liquid water content, and remove the statements that directly attribute observed differences to particle acidity. The abstract and summary (and ideally title) should only provide conclusive statements about what can be directly supported by the data, which does not include effects of particle acidity alone. Further discussion about the potential role of phase state alone would also help; the added paragraph at the end of Section 3.2.3 is useful for this purpose, but remains

vague and unsupported by evidence (how do we know that the LWC equilibration time is "probably" on the order of minutes? Citations here would help).

Authors reply:

We thank the Editor for pointing this out. We feel that our response to the previous comments (see above) by the Editor highlights clearly the comments raised by the Editor here. We further added the following sentences to page 31, line 25 and reads:

"The observed differences between the acidified (ER444) and non-acidified (ER666) seed aerosol experiments may be due to the acidity of the seed aerosol, liquid water content, and/or the phase state of the aerosol particles. Furthermore, more work is needed to gain insight into mechanisms of reactant formation and partitioning to deliquescent particles and to particles lacking the aqueous phase."

We prepared the acidic and non-acidic seed aerosol with the method that has been used in chamber experiments for decades. In our new revised manuscript, we feel strongly that our new terminology "acidified seed aerosol" and "non-acidified seed aerosol" reflects the experimental conditions used in this study. We apologies for any misunderstanding associated with the terms "acidic" and "non-acidic".

To reflect the editor concerns, we replaced the last paragraph in section 3.2.3 that reads:

"As stated above, the non-acidic seed aerosols were dry based on E-AIM modelling. This may be a major difference in the acidic and non-acidic conditions and probably one of the main drivers of differences in the chemistry between the two systems. Note that E-AIM is an equilibrium model, which has its own limitations. Therefore, it is important to consider that under our experimental conditions, the system should be at chemical equilibrium, which can be difficult to characterize. In our experimental system, we are going from liquid solution in the nebulizer reservoir, to ammonium sulfate particles in near -100% RH air exiting the nebulizer to chamber dry air in less than a minute. Then the particles have a chance to equilibrate in the chamber for a matter of hours. The equilibrium time needed for acidic wet aerosols to adjust their LWC to the chamber RH conditions is probably on the order of minutes, so they are probably close to their E-AIM predicted LWCs in the chamber. However, the time needed for the wet neutral aerosols to completely dry out at room temperature may be on the order of hours. It is probable that under our experimental conditions the wet neutral aerosol do not reach the E-AIM predicted dryness on the time scale of the chamber residence time. This may explain why the two systems are not drastically different chemically as expected."

With the following paragraph (last paragraph in section 3.2.3 in the re-revised manuscript and reads:

"The preparation of non-acidified and acidified seed aerosol by nebulizing aqueous solutions of $NH_4SO_4$ and ($NH_4SO_4 + H_2SO_4$), respectively, has been used in chamber experiments for many years

(Czoschke et al., 2003; Deng et al., 2021; Zhang et al., 2023). As stated above, the LWC and acidity of the aerosol were estimated using E-AIM modeling, which assumes the vapor-particle partitioning equilibrium in the system. The non-acidified seed aerosols were dry based on E-AIM modelling. This may be a major difference in the acidified and non-acidified conditions and probably one of the main drivers of differences in the chemistry between the two systems. Note that E-AIM is an equilibrium model, which has its own limitations. Therefore, it is important to consider that under our experimental conditions, the system should be at chemical equilibrium, which can be difficult to characterize. In our experimental system, we are going from liquid solution in the nebulizer reservoir, to ammonium sulfate particles in near -100% RH air exiting the nebulizer to chamber dry air in less than a minute. Then the particles have a chance to equilibrate in the chamber for a matter of hours. The equilibrium time needed for acidified wet aerosols to adjust their LWC to the chamber RH conditions is probably on the order of minutes to ten of minutes (Saleh et al. 2013), so they are probably close to their E-AIM predicted LWCs in the chamber. However, the time needed for the wet neutral aerosols to completely dry out at room temperature may be on the order of hours. It is probable that under our experimental conditions the wet neutral aerosol do not reach the E-AIM predicted dryness on the time scale of the chamber residence time. This may explain why the two systems are not drastically different chemically as expected."

The following reference were added to the re-revised manuscript.

Czoschke, N. M., Jang, M., and Kamens, R. M.: Effect of acidic seed on biogenic secondary organic aerosol growth, Atmos Environ, 37, 4287-4299, http://dx.doi.org/10.1016/S1352-2310(03)00511-9, 2003.

Deng, Y., Inomata, S., Sato, K., Ramasamy, S., Morino, Y., Enami, S., and Tanimoto, H.: Temperature and acidity dependence of secondary organic aerosol formation from α-pinene ozonolysis with a compact chamber system, Atmos Chem Phys, 21, 5983-6003,

https://doi.org/10.5194/acp-21-5983-2021, 2021.

Saleh, R., Donahue, N. M., and Robinson, A. L.: Time Scales for Gas-Particle Partitioning Equilibration of Secondary Organic Aerosol Formed from Alpha-Pinene Ozonolysis, Environmental Science & Technology, 47, 5588-5594, https://doi.org/10.1021/es400078d, 2013.

Zhang, J., Shrivastava, M., Zelenyuk, A., Zaveri, R. A., Surratt, J. D., Riva, M., Bell, D., and Glasius, M.: Observationally Constrained Modeling of the Reactive Uptake of Isoprene-Derived Epoxydiols under Elevated Relative Humidity and Varying Acidity of Seed Aerosol Conditions, ACS Earth and Space Chemistry, 7, 788-799, https://doi.org/10.1021/es400078d, 2023.

5. (If, as the authors suggest in one reviewer response, this is all just a misunderstanding because "acidic" vs. "non-acidic" refers to the initial atomizer solution and not the particles, that needs to be very well clarified throughout. The authors should then use different descriptors to separate the two experiments, because the current terminology is misleading, as it strongly implies that particle acidity (not atomizer solution acidity) is the factor on which subsequent organic aerosol composition differences depend).

Authors reply:

This comment is clearly clarified in the re-revised manuscript (see our response to the previous comments of the Editor above).

6. Like the reviewers, I believe this manuscript will be a valuable addition to the literature once it is revised such that its conclusions are better supported by the evidence provided throughout, and I hope the authors agree that this will improve the manuscript's clarity and usefulness. I look forward to reading a new revision.

Authors reply:

We greatly appreciate the Editor for his valuable input and comments. We feel that the conclusions in the re-revised manuscript are supported by the evidence provided throughout the manuscript.